# Defective STIM-mediated store operated Ca²⁺ entry in hepatocytes leads to metabolic dysfunction in obesity

Ana Paula Arruda[1†], Benedicte Mengel Pers[1†‡], Günes Parlakgul[1], Ekin Güney[1], Ted Goh[1], Erika Cagampan[1], Grace Yankun Lee[1], Renata L Goncalves[1], Gökhan S Hotamisligil[1,2]*

[1]Department of Genetics and Complex Diseases, Sabri Ülker Center, Harvard TH Chan School of Public Health, Boston, United States; [2]Broad Institute of MIT and Harvard, Cambridge, United States

**Abstract** Defective Ca²⁺ handling is a key mechanism underlying hepatic endoplasmic reticulum (ER) dysfunction in obesity. ER Ca²⁺ level is in part monitored by the store-operated Ca²⁺ entry (SOCE) system, an adaptive mechanism that senses ER luminal Ca²⁺ concentrations through the STIM proteins and facilitates import of the ion from the extracellular space. Here, we show that hepatocytes from obese mice displayed significantly diminished SOCE as a result of impaired STIM1 translocation, which was associated with aberrant STIM1 O-GlycNAcylation. Primary hepatocytes deficient in STIM1 exhibited elevated cellular stress as well as impaired insulin action, increased glucose production and lipid droplet accumulation. Additionally, mice with acute liver deletion of STIM1 displayed systemic glucose intolerance. Conversely, over-expression of STIM1 in obese mice led to increased SOCE, which was sufficient to improve systemic glucose tolerance. These findings demonstrate that SOCE is an important mechanism for healthy hepatic Ca²⁺ balance and systemic metabolic control.

DOI: https://doi.org/10.7554/eLife.29968.001

*For correspondence: ghotamis@hsph.harvard.edu

†These authors contributed equally to this work

Present address: ‡The Niels Bohr Institute, University of Copenhagen, Copenhagen, Denmark

Competing interests: The authors declare that no competing interests exist.

## Introduction

The endoplasmic reticulum (ER) is a key cellular organelle coordinating a variety of essential processes such as protein synthesis and secretion, lipid biosynthesis, glucose metabolism and redox reactions. Additionally, ER is the main site of Ca²⁺ storage in the cell (*Gardner et al., 2013*; *Hotamisligil, 2010*). Structurally, ER comprises a complex network of membranes spread throughout the cytoplasm, which establishes physical and functional contacts with many other organelles including mitochondria, endosomes, lipid droplets and the plasma membrane (*Lynes and Simmen, 2011*; *Phillips and Voeltz, 2016*).

Given its critical role in the cell, the functionality of the ER is tightly monitored by proteins that can communicate stress signals to other proteins or compartments of the cell in order to restore ER function. The most well-known adaptive pathway in the ER is the unfolded protein response (UPR), which has the general goal of restoring ER function by enhancing ER folding capacity, decreasing protein translation and increasing protein degradation (*Gardner et al., 2013*; *Hetz et al., 2015*; *Hotamisligil, 2010*; *Wang and Kaufman, 2016*). Over the past decade it has been widely documented that chronic stress imposed by excess nutrients and energy leads to ER dysfunction and UPR activation in a variety of metabolic tissues in both mouse models and obese humans, which is associated with cellular stress, inflammation, and metabolic dysfunction (*Boden et al., 2008*; *Gregor et al., 2009*; *Hotamisligil, 2010*; *Nakatani et al., 2005*; *Ozcan et al., 2004*; *Sharma et al., 2008*; *Wang and Kaufman, 2016*). Alleviation of ER stress by chemical or molecular chaperones

**eLife digest** Obesity is a chronic metabolic disorder. Some people's genetics make them more vulnerable to the condition, and it is generally caused by eating too much and moving too little. The resulting surplus of nutrients affects the cells and organs of the body in several adverse ways. For example, excessive nutrients impair a compartment within cells called the endoplasmic reticulum. This compartment is where many proteins and fats are made and transported. It is also the site for a lot of metabolic processes, and the main place in the cell where calcium ions are stored. Many proteins need calcium ions to work properly, including metabolic enzymes. In obesity, the endoplasmic reticulum becomes less able to store calcium ions.

A protein called STIM1 senses and regulates the levels of calcium ions in the endoplasmic reticulum. When calcium levels drop, STIM1 moves along the endoplasmic reticulum membrane towards the part that is next to the cell surface. Here, STIM1 joins up with a calcium channel called Orai1. The STIM1-Orai1 complex allows calcium ions to enter the cell and replenish its levels in the endoplasmic reticulum.

Arruda, Pers et al. have now asked if STIM1 is altered in obesity and, if so, whether it contributes to the endoplasmic reticulum's inability to maintain proper calcium levels. High-resolution microscopy and biochemical techniques confirmed that STIM1 is indeed compromised in liver cells from obese mice. In these cells, STIM1 was found in unusual small clusters. It also could not move along the endoplasmic reticulum membrane when calcium levels dropped. As a result of these navigational errors, STIM1 failed to couple with Orai1, meaning less calcium could enter the cell. Further work identified that a small sugar molecule that is added onto STIM1 in obesity is behind its reduced ability to move accurately.

Arruda, Pers et al. next created mice that lacked STIM1 in their liver. These mice showed signs of metabolic abnormalities. Notably, when STIM1 levels were experimentally increased in obese mice, it restored calcium levels in the endoplasmic reticulum closer to normal, and improved metabolism too.

Thus, regulating calcium levels in the endoplasmic reticulum via proteins such as STIM1 is essential for maintaining a healthy metabolism. Interventions to correct calcium levels may have therapeutic promise to combat metabolic diseases.

DOI: https://doi.org/10.7554/eLife.29968.002

dramatically improves metabolic control and insulin sensitivity and reduces inflammation in mouse models of obesity (*Fu et al., 2015*; *Kammoun et al., 2009*; *Ozawa et al., 2005*; *Ozcan et al., 2006*) as well as in humans (*Kars et al., 2010*; *Xiao et al., 2011*).

More recently, it has become clear that $Ca^{2+}$ storage in the ER is compromised in the setting of obesity and other metabolic diseases (*Arruda and Hotamisligil, 2015*; *Ozcan and Tabas, 2016*). $Ca^{2+}$ in the ER is essential for chaperone-mediated protein folding and secretion, as well as for the function of metabolic enzymes. $Ca^{2+}$ concentration in the ER is tightly regulated by coordinated action between SERCA, which pumps $Ca^{2+}$ from the cytosol into the ER lumen, and the $IP_3$ receptor (IP3R) or Ryanodine receptor, which release $Ca^{2+}$ from the ER into the cytosol (*Clapham, 2007*). In obesity, hepatic SERCA activity is compromised (*Fu et al., 2011*; *Meikle and Summers, 2017*; *Rong et al., 2013*) while the activity of the IP3R1 $Ca^{2+}$ channel is increased (*Arruda et al., 2014*; *Feriod et al., 2017*; *Wang et al., 2012*). These alterations result in decreased ER $Ca^{2+}$ content, loss of folding capacity, ER stress, inflammation, impaired insulin action and abnormal glucose metabolism (*Arruda and Hotamisligil, 2015*). Additionally, in this setting a 'leaky' ER contributes to both higher cytosolic $Ca^{2+}$ and mitochondrial $Ca^{2+}$ uptake, which has important implications for cytosolic $Ca^{2+}$ signaling and mitochondrial dysfunction seen in metabolic diseases (*Arruda et al., 2014*; *Feriod et al., 2017*; *Ozcan et al., 2013*; *Wang et al., 2012*; *Xiao et al., 2011*). Accordingly, strategies to restore hepatic ER $Ca^{2+}$ levels, such as overexpression of SERCA or suppression of IP3R, improve ER function and promote metabolic homeostasis in mouse models of obesity (*Arruda et al., 2014*; *Feriod et al., 2017*; *Fu et al., 2011*; *Wang et al., 2012*). In a similar fashion, administration of compounds that increase ER $Ca^{2+}$ content and improve ER function, e.g. SERCA

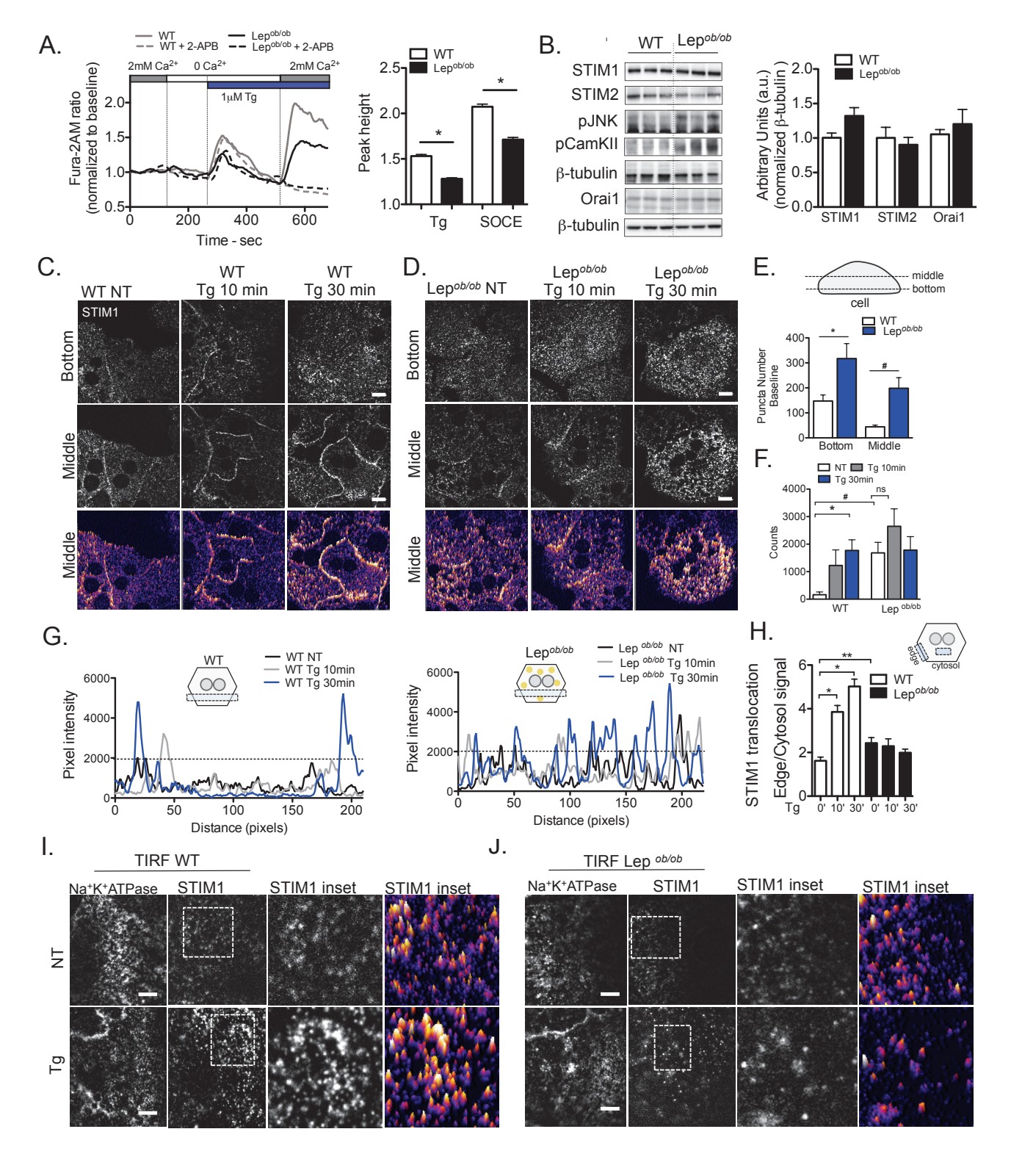

**Figure 1.** Obesity leads to decreased SOCE and impaired STIM1 translocation in primary hepatocytes. (**A**) Left panel: representative Fura-2AM based cytosolic $Ca^{2+}$ measurements in primary hepatocytes isolated from wild type (WT) and leptin-deficient ($Lep^{ob/ob}$) obese mice. ER $Ca^{2+}$ stores were depleted with 1 µM thapsgargin (Tg), a SERCA inhibitor. SOCE was evaluated by substituting the extracellular media (0 mM $Ca^{2+}$) with media containing 2 mM $Ca^{2+}$. Dashed lines show $Ca^{2+}$ measurements in the presence of 50 µM 2-Aminoethoxydiphenyl borate (2-APB). Right panel:

*Figure 1 continued on next page*

*Figure 1 continued*

quantification of Tg-induced $Ca^{2+}$ release (reflecting ER $Ca^{2+}$ content) and SOCE based on the measurements shown in (**A**), n = 160 WT and n = 250 *Lep^{ob/ob}* cells for Tg response and n = 234 WT and n = 303 *Lep^{ob/ob}* cells for SOCE response. Data were pooled across six independent experiments. *p<0.0001 (**B**) Left panel: Immunoblot analysis of protein expression levels in total liver lysate from WT and *Lep^{ob/ob}* mice, Right panel: quantification of the western blots n = 4, representative of 4–5 experiments. (**C and D**) Confocal images of immunofluorescence staining for endogenous STIM1 in primary hepatocytes from WT and *Lep^{ob/ob}* animals, treated with DMSO (vehicle, NT) or 1 µM Tg for 10 and 30 min. NT refers to 'not Tg treated' (**E**) Quantification of STIM1 puncta/cluster number in the bottom and middle cross-section of non-treated (DMSO) hepatocytes from WT and *Lep^{ob/ob}* animals.n = 5–6 fields (WT) and 4–5 fields (*Lep^{ob/ob}*), representative of 4 independent experiments, *p=0.02 #p=0.003 (**F**) Quantification of pixel intensity in a set area (125 × 125 pixels) of the middle section of the cells from WT and *Lep^{ob/ob}* animals, treated with 1 µM Tg or vehicle (quantification methods depicted in *Figure 1—figure supplement 1 F*), n = 4–9 cells, representative of 4 independent experiments, *p=0.02 #p=0.008 (**G**) Representative profile plots of STIM1 levels (pixel intensities) in a defined area (box) across cells treated with DMSO or 1 µM Tg for 10 or 30 min. Left: cells from WT animals, right: cells from *Lep^{ob/ob}* animals. (**H**) Quantification of STIM1 translocation by calculating the ratio between the mean STIM1 pixel intensity at a selected area of the edge of the cell relative to the same measurement performed in the cytosolic area near the edge of the cell, n = 3–4 ratios per cell, quantified in 2–8 cells for each condition *p<0.0001 **p=0.009 (**I and J**) Representative TIRF images of STIM1 and $Na^{+}K^{+}$-ATPase (PM marker) in cells from WT (**I**) and *Lep^{ob/ob}* mice (**J**) treated with 1 µM Tg or vehicle for 10 min. NT refers to "not Tg treated. For all graphs, error bars denote s.e.m. Scale: 10 µm.

DOI: https://doi.org/10.7554/eLife.29968.003

The following source data and figure supplements are available for figure 1:

**Source data 1.** Source data for *Figure 1*.
DOI: https://doi.org/10.7554/eLife.29968.004

**Figure supplement 1.** Obesity leads to impaired STIM1 translocation in primary hepatocytes.
DOI: https://doi.org/10.7554/eLife.29968.005

**Figure supplement 1—source data 1.** Source data for *Figure 1—figure supplement 1*.
DOI: https://doi.org/10.7554/eLife.29968.006

**Figure supplement 2.** Decline of STIM1 trafficking in primary hepatocytes over the course of high fat diet (HFD)
DOI: https://doi.org/10.7554/eLife.29968.007

**Figure supplement 2—source data 1.** Source data for *Figure 1—figure supplement 2*.
DOI: https://doi.org/10.7554/eLife.29968.008

**Figure supplement 3.** Confocal images of immunofluorescence staining of endogenous STIM2 in primary hepatocytes from WT and *Lep^{ob/ob}* animals, treated with or without 1 µM Tg for 10 min.
DOI: https://doi.org/10.7554/eLife.29968.009

**Figure supplement 3—source data 1.** Source data for *Figure 1—figure supplement 3*.
DOI: https://doi.org/10.7554/eLife.29968.010

agonists (*Kang et al., 2016*), IP3R antagonists (*Ozcan et al., 2012*), or azoromide (*Fu et al., 2015*), improves metabolic health in obese mice.

In addition to SERCA and IP3Rs, $Ca^{2+}$ homeostasis in the ER is sensed and regulated by a third major adaptive/homeostatic system orchestrated by the STIM-Orai complex. STIM proteins (STIM1 and STIM2) are found in the ER and sense luminal $Ca^{2+}$ through N-terminal $Ca^{2+}$ binding EF hand domain. In the resting state, STIM proteins are bound to $Ca^{2+}$ and spread evenly throughout the ER membrane. Upon ER $Ca^{2+}$ release, STIM forms oligomers (seen in confocal microscopy as puncta), which translocate to junctions between ER and plasma membrane (PM), where they couple with the PM channel protein Orai1. This coupling results in the import of $Ca^{2+}$ from the extracellular compartment to the cytosol, providing spatial $Ca^{2+}$ signals that then influence cellular signaling and promote $Ca^{2+}$ replenishment into the ER lumen through SERCA, in the process known as store operated $Ca^{2+}$ entry or SOCE (*Feske, 2007*; *Prakriya and Lewis, 2015*; *Soboloff et al., 2012*; *Wu et al., 2007*).

Given that obesity leads to impaired ER $Ca^{2+}$ handling and this is a key mechanism associated with ER dysfunction in the obese condition, we examined whether alterations in SOCE system exist and contribute to the inability of ER to restore and or maintain $Ca^{2+}$ levels. Here we show that STIM1 is modified and SOCE is defective in hepatocytes from obese mice due to the inability of STIM1 to translocate and couple with Orai1 at the PM. Hepatocytes lacking STIM1 displayed stress, increased glucose production and insulin resistance, whereas over-expression of STIM1 in the liver of obese mice significantly improved glucose intolerance. These findings support a critical role for STIM1-mediated SOCE in the maintenance of healthy $Ca^{2+}$ balance in the ER, insulin action, and systemic glucose metabolism.

# Results

In order to determine the impact of obesity on SOCE in the liver, we isolated primary hepatocytes from lean wild-type (WT) and genetically obese ($Lep^{ob/ob}$) mice and measured $Ca^{2+}$ influx through STIM/Orai1 using the ratiometric calcium dye, Fura-2AM (*Poenie and Tsien, 1986*). First, the cells were incubated in $Ca^{2+}$-free medium, and ER $Ca^{2+}$ store depletion was induced by the addition of the SERCA inhibitor thapsigargin (Tg). In agreement with earlier reports (*Arruda et al., 2014*) the initial rise in cytosolic $Ca^{2+}$ induced by Tg, which reflects the ER $Ca^{2+}$ content, was significantly lower in hepatocytes isolated from obese animals compared with WT cells (*Figure 1A*). Next, we induced SOCE through STIM/Orai1 by substituting the $Ca^{2+}$-free media with a media containing 2 mM $Ca^{2+}$. As depicted in *Figure 1A*, the $Ca^{2+}$ entry in primary hepatocytes from obese mice was markedly reduced relative to control cells. To further confirm that the observed $Ca^{2+}$ entry was mediated by SOCE, we added the SOCE inhibitor 2-apb, which completely blocked $Ca^{2+}$ influx in both genotypes (*Figure 1A*, dotted lines). This finding is consistent with a report of impaired SOCE in the steatotic hepatocytes from Zucker rats (*Wilson et al., 2015*).

As we observed impaired SOCE in hepatocytes from obese animals we next evaluated whether the expression levels of the main SOCE components were altered in obesity. As shown in *Figure 1—figure supplement 1A*, we did not detect differences in expression of Orai1 in the livers of mice with genetic or high-fat diet (HFD)-induced obesity. Despite a modest increase in the mRNA levels of stim1 and stim2, protein levels of STIM1, STIM2 and Orai1 remained unchanged in the livers of mice with both genetic and diet induced obesity (*Figure 1B* and *Figure 1—figure supplement 1B*). However, phospho-JNK (pJNK), a marker of inflammatory stress (*Hirosumi et al., 2002*) and phospho-Calmodulin kinase (pCaMKII), a marker of elevated cytosolic $Ca^{2+}$ (*Ozcan et al., 2012*), were increased in the livers of obese animals (*Figure 1B* and *Figure 1—figure supplement 1B*). Thus decreased $Ca^{2+}$ import through SOCE in hepatocytes derived from obese mice appears to be independent of the expression levels of critical SOCE components.

We therefore asked whether the decreased $Ca^{2+}$ import through SOCE in hepatocytes from obese mice was a result of a functional alteration in STIM1 translocation in the ER membrane. To evaluate the activity of STIM1 proteins in obesity, we first validated antibodies for endogenous immunostaining (*Figure 1*; *Figure 1—figure supplement 1C*). Next, we isolated primary hepatocytes from lean and obese mice, induced $Ca^{2+}$ store depletion with Tg and determined STIM1 localization by immunostaining. As shown in *Figure 1C and D* and *Figure 1—figure supplement 1D*, while STIM1 was evenly distributed in the ER membrane of resting cells, Tg treatment led to a dramatic translocation to areas of the ER membrane in close proximity with the PM. This effect was observed within 5–10 min of Tg treatment and persisted for 30 min. However, in cells from obese ($Lep^{ob/ob}$) mice, STIM1 distribution at baseline was punctate throughout the entire ER (*Figure 1D* and quantified in *Figure 1E*), and its translocation to areas of ER/PM junction in response to Tg was dramatically impaired. As shown in *Figure 1F* and *Figure 1—figure supplement 1E*, in WT cells, Tg treatment increased the number of high-intensity pixels, indicating the formation of the punctate structures, which represent oligomerization of STIM proteins. In contrast, hepatocytes derived from obese mice displayed higher intensity pixel counts at baseline with no further increase following Tg treatment. Defective STIM1 translocation in hepatocytes from obese animals is also illustrated in the line graphs showing the representative intensity profile of individual cells before and after Tg treatment (*Figure 1G* and also see *Figure 1—figure supplement 1F* for image analysis details). In WT cells, Tg treatment increased the signal intensity at the edge of the cell and decreased it in the cytosol, quantified as the STIM1 ratio between these compartments (*Figure 1H*). In hepatocytes from obese mice, the average signal intensity at baseline was higher than in WT cells, and did not change significantly in response to Tg (*Figure 1H*), indicating that STIM1 translocation is markedly defective in these cells.

To examine STIM1 translocation after Tg stimulation in more detail and using an alternative analytical tool, we performed total internal reflection fluorescence (TIRF) microscopy, in which fluorophores in the vicinity of the cell surface are selectively excited. As shown in *Figure 1I*, in WT cells Tg treatment induced STIM1 puncta formation and accumulation in areas close to the PM. This response was diminished in cells derived from obese animals (*Figure 1J*). In these experiments, $Na^{+}K^{+}$ ATPase was used as a constitutive marker of PM and its staining pattern did not change in the presence of Tg.

We also evaluated the time course of the defect in STIM1 trafficking during the development of obesity in mice. As shown in *Figure 1—figure supplement 2A* and *Figure 1—figure supplement 2B*, after 3 and 5 weeks of HFD, STIM1 showed some degree of translocation towards the plasma membrane in resting cells, indicating that short term HFD exposure is sufficient to trigger ER $Ca^{2+}$ release, the driving force for STIM1 translocation. Additionally, at 3 weeks HFD, STIM1 translocation stimulated by Tg was not significantly impaired compared with cells derived from chow fed animals. However, the defect in STIM1 translocation induced by Tg was apparent in cells isolated from mice fed a HFD for 5 weeks and was more pronounced after 7 and 11 weeks of HFD (*Figure 1—figure supplement 2A* and *Figure 1—figure supplement 2B*). These data support that defective STIM1 translocation may be a common feature of obesity in independent experimental models (*Lep$^{ob/ob}$* and HFD) and that this phenotype occurs independent of blood glucose levels and prior to the emergence of marked hyper-insulinemia and insulin resistance found in obesity (*Figure 1—figure supplement 2C*).

Next, we examined the activation and translocation of STIM2 in cells from lean (WT) and obese (*Lep$^{ob/ob}$*) mice. The role of STIM2 in the regulation of SOCE has been explored only recently, and its activation seems to result from smaller fluctuations in $Ca^{2+}$ due to its lower affinity for the ion, while STIM1 is activated when the extent of ER depletion increases (*Berna-Erro et al., 2017*; *Oh-Hora et al., 2008*; *Prakriya and Lewis, 2015*). In hepatocytes, STIM2 is expressed at a lower level than STIM1 (*López et al., 2012*; *Williams et al., 2001*), however it was detectable both by staining and by western blot analysis. We did not observe aberrant STIM2 puncta formation in cells from *Lep$^{ob/ob}$* mice but a tendency for translocation at baseline. Upon Tg-induced store depletion, we observed a marked STIM2 translocation to areas close to the PM in both WT cells and *Lep$^{ob/ob}$* cells, however the degree of translocation in *Lep$^{ob/ob}$* cells was less pronounced than in WT cells (*Figure 1—figure supplement 3A*). Thus, the obesity-related defect in STIM2 translocation was present but less dramatic than that observed for STIM1. Overall, these findings suggest that defective SOCE in primary hepatocytes from obese mice is predominantly related to aberrant STIM1 puncta formation and inefficient STIM1 re-localization from the bulk ER membrane to ER/PM junction areas after ER Ca2+ store depletion.

In order to gain insight into mechanisms underlying this phenomenon, we considered the possibility that altered STIM1 post-translational modification could be involved in its defective translocation capacity in the context of obesity. Two known post-translational modifications of STIM1 which influence its trafficking are phosphorylation and O-GlcNAcylation (*Pozo-Guisado et al., 2013*; *Zhu-Mauldin et al., 2012*). Phosphorylation of STIM1 at Ser621 and Ser575 regulate its interaction with the microtubule plus-end-tracking protein EB1, enabling STIM1 to move in the ER membrane (*Pozo-Guisado et al., 2013*). Therefore, we asked whether a reduction in STIM1 phosphorylation at these sites may explain its defective trafficking in obesity. Surprisingly however, we found that phosphorylation of STIM1 at Ser621 and Ser575 was actually increased in liver lysates from obese mice (*Figure 2A*). This indicates that lack of phosphorylation at these residues does not underlie the defective translocation of STIM1, and suggests that hepatic STIM1 may be released from EB1 in the basal state, potentially as a response to the decreased ER $Ca^{2+}$ level observed in hepatocytes from obese mice.

STIM1 can also be modified by O-linked N-acetyl glucosamine (O-GlcNAc), a post-translational modification of serine or threonine amino acids. Previous work has shown that STIM1 O-GlcNAcylation impairs the ability of the protein to move in the ER membrane and to form punctate structures (*Zhu-Mauldin et al., 2012*). Additionally, conditions of nutrient and substrate excess, including obesity, lead to increase in cellular O-GlcNAcylation levels (*Vosseller et al., 2002*; *Yang and Qian, 2017*; *Dentin et al., 2008*; *Yang et al., 2008*). These studies indicate the possibility that metabolic stress may interfere with SOCE via O-GlcNAcylation of STIM1, which disrupts its proper trafficking. To test this hypothesis, we first examined global O-GlcNAcylation in primary hepatocytes from WT and obese mice. As shown in *Figure 2B*, and in agreement with previous observations (*Baldini et al., 2016*), global O-GlcNAcylation was higher in hepatocytes derived from obese animals compared with their controls. This effect was amplified by treatment with PugNac, a specific inhibitor of O-GlycNAcase (OGA), the enzyme that catalyzes removal of OglcNAc sugars from proteins. Next, we performed immunoprecipitation of O-GlcNAc-modified proteins using an OglcNAc-specific antibody from hepatocyte lysates. We found that STIM1 was present among the OglcNAcylated proteins and it was more abundant in cells from obese mice (*Figure 2C*). As a complementary approach,

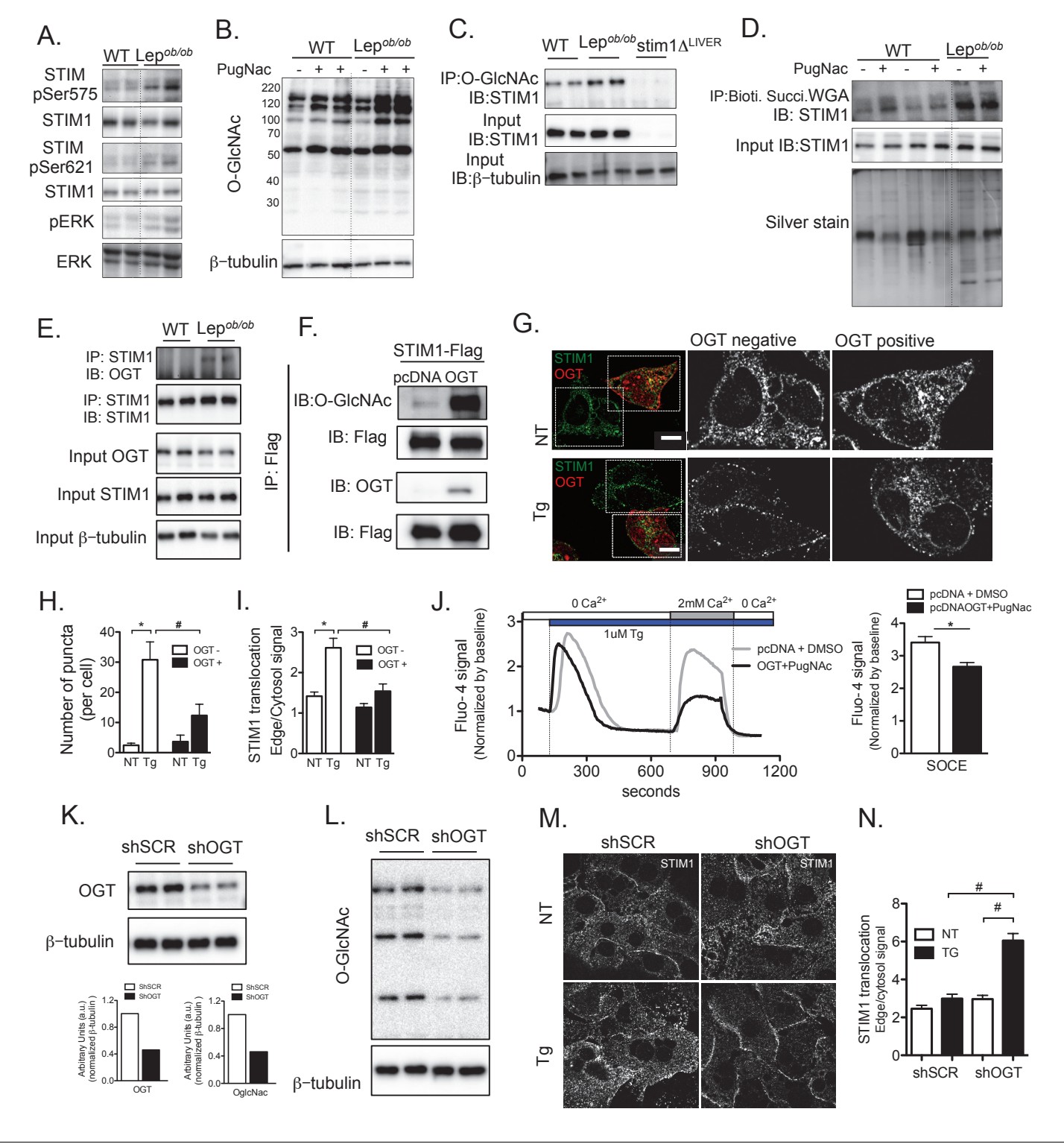

**Figure 2.** Post-translational modification of STIM1 by phosphorylation and O-GlcNAcylation in obesity. (**A**) Immunoblot analysis of the phosphorylation level of STIM1 at Ser575 and Ser621 and of ERK phosphorylation in total liver lysates from WT and *Lep^ob/ob* animals (**B**) Immunoblot analysis of O-GlcNAcylation of total lysates extracted from primary hepatocytes derived from WT and *Lep^ob/ob* animals in the presence or absence of overnight treatment with 10 μM of PugNac, an inhibitor of O-GlcNAcase. (**C**) Immunoprecipitation of O-GlcNAcylated proteins derived from WT and *Lep^ob/ob* hepatocytes followed by immunoblotting for STIM1. Cells from STIM1-deficient mice (STIM1Δ^Liver) were used as a negative control. (**D**) Total lysates from WT or *Lep^ob/ob* hepatocytes treated or not with 10 μM of PugNac overnight were incubated with biotinylated, succinylated WGA and precipitated

*Figure 2 continued on next page*

Figure 2 continued

with streptavidin-conjugated beads. The recovered proteins were used for immunoblotting for STIM1. An aliquot of each precipitate was run in a gel and silver stained. (E) Immunoprecipitation of STIM1 in total lysates from primary hepatocytes derived from WT and $Lep^{ob/ob}$ animals followed by immunoblotting for OGT. Input for OGT and STIM1 in the total lysates prior to immunoprecipitation is shown (F) Immunoprecipitation of STIM1-Flag from total lysates stably expressing STIM-Flag in the presence or absence of OGT followed by immunoblotting for O-GlcNac and OGT. (G) Confocal images of endogenous immunofluorescent staining of STIM1 in Hepa1-6 cells transfected with OGT, treated with DMSO or 1 µM Tg stimulation. Non-transfected cells present in the same dish were used as controls for quantification. OGT expressing cells were identified by the presence of RFP expression. NT refers to 'not Tg treated' (H) Number of STIM1 puncta quantified per cell, n = 7–14 cells per group, representative of 3 independent experiments, *p=0.0003 #p=0.03. (I) Quantification of STIM1 translocation by calculating the ratio between STIM1 protein signal at the edge of cell relative to the cytosol close by. For OGT n = 60–64 areas and for OGT +n = 22–30 areas per group, representative of 3 independent experiments, *p<0.0001 #p=0.004 (J) Left panel: representative Fura-2AM based cytosolic $Ca^{2+}$ measurements in Hepa1-6 cells overexpressing pcDNA control or OGT. Right panel: quantification of $Ca^{2+}$ influx through SOCE. n = 51 (pcDNAt +DMSO) n = 61 (OGT +PugNAc) *p=0.0007. (K and L) Immunoblot analysis and densitometric quantification of OGT and OglcNAc expression in primary hepatocytes from $Lep^{ob/ob}$ animals infected with adenovirus expressing scrambled shRNA (shSCR) and shRNA against OGT (shOGT) for 48 hr. (M) Confocal images of immunofluorescence staining for endogenous STIM1 in primary hepatocytes from $Lep^{ob/ob}$ animals expressing scrambled shRNA (shSCR) and shRNA against OGT (shOGT) for 48 hr. NT refers to 'not Tg treated' (N) Quantification of STIM1 translocation by calculating the ratio between the mean STIM1 pixel intensity at a selected area of the edge of the cell relative to the same measurement performed in the cytosolic area near the edge of the cell, n = 3 ratios per cell, 5 cells per field in four fields per condition,*p=0.04 # p<0.001.For all graphs, error bars represent s.e.m. Scale: 10 µm.

DOI: https://doi.org/10.7554/eLife.29968.011

The following figure supplement is available for figure 2:

**Figure supplement 1.** STIM1 O-GlcNAcylation in primary hepatocytes and Hepa 1-6 cells.
DOI: https://doi.org/10.7554/eLife.29968.012

we utilized biotinylated-succinylated wheat germ agglutinin (succinylated-WGA), a lectin that preferentially binds N-acetylglucosamine versus other sugars (*Baldini et al., 2016*; *Hu et al., 2010*). Following precipitation with streptavidin-conjugated magnetic beads, STIM1 was detected in the pool of proteins modified by O-GlcNac at higher levels in samples derived from obese animals relative to WT controls (*Figure 2D*). Additionally, as shown in *Figure 2E*, OGT precipitates with STIM1 in hepatocytes derived from obese animals, indicating that OGT and STIM1 exist in a complex in obesity, consistent with higher levels of O-GlcNac-modified STIM1 in this condition. To explore whether STIM1 is also modulated by OglcNAcylation in the HFD context and study the time course of this modification, we isolated hepatocytes from animals fed a HFD for 3, 5 and 11 wks and pull down STIM1 using succinylated-WGA. As shown in *Figure 2—figure supplement 1A*, STIM1 was progressively modified by O-GlcNac in this context starting at 3 weeks on HFD.

In order to test if O-GlcNAc modification may alter STIM1 function in hepatocytes, we first overexpressed OGT in Hepa1-6 cells. As expected, overexpression of OGT strongly increased global protein modification with O-GlcNac (*Figure 2—figure supplement 1B*). We then co-expressed OGT and STIM1 tagged with a Flag peptide (*Figure 2—figure supplement 1C*) to determine whether STIM1 can be directly O-GlcNacylated by OGT in this system. Immunoprecipitation of STIM1-Flag with a Flag specific antibody demonstrated its enhanced O-GlcNAc modification in OGT overexpressing cells (*Figure 2F*). Additionally, STIM1 is able to directly bind to OGT in this cell model (*Figure 2F*), similar to what we observed in hepatocytes derived from obese animals (*Figure 2E*). Having established this cellular system with enhanced STIM1 O-GlcNacylation, we then assessed its translocation in cells transfected with OGT labeled with RFP, using un-transfected cells in the same culture plate as internal controls for quantification (*Figure 2G*). These experiments showed that STIM1 puncta formation and translocation following Tg treatment was markedly impaired in cells overexpressing OGT compared with control cells (*Figure 2G,H and I*). Additionally, overexpression of OGT led to a significant impairment in SOCE compared with the controls (*Figure 2J*), with no significant differences in the response to Tg (data not shown).

Next, to determine whether inhibition of the O-GlcNac modification could rescue STIM1 translocation defect in $Lep^{ob/ob}$ cells, we used adenoviral gene delivery to express an shRNA targeting OGT to down-regulate OGT and thus the O-GlcNacylation capacity of the cell. As can be seen in *Figure 2K and L*, the expression of OGT shRNA lead to a 60% decrease in its expression and activity. As shown in *Figure 2M and N*, down-regulation of OGT resulted in increased STIM1 translocation capacity compared with cells expressing a scrambled shRNA. Taken together these data

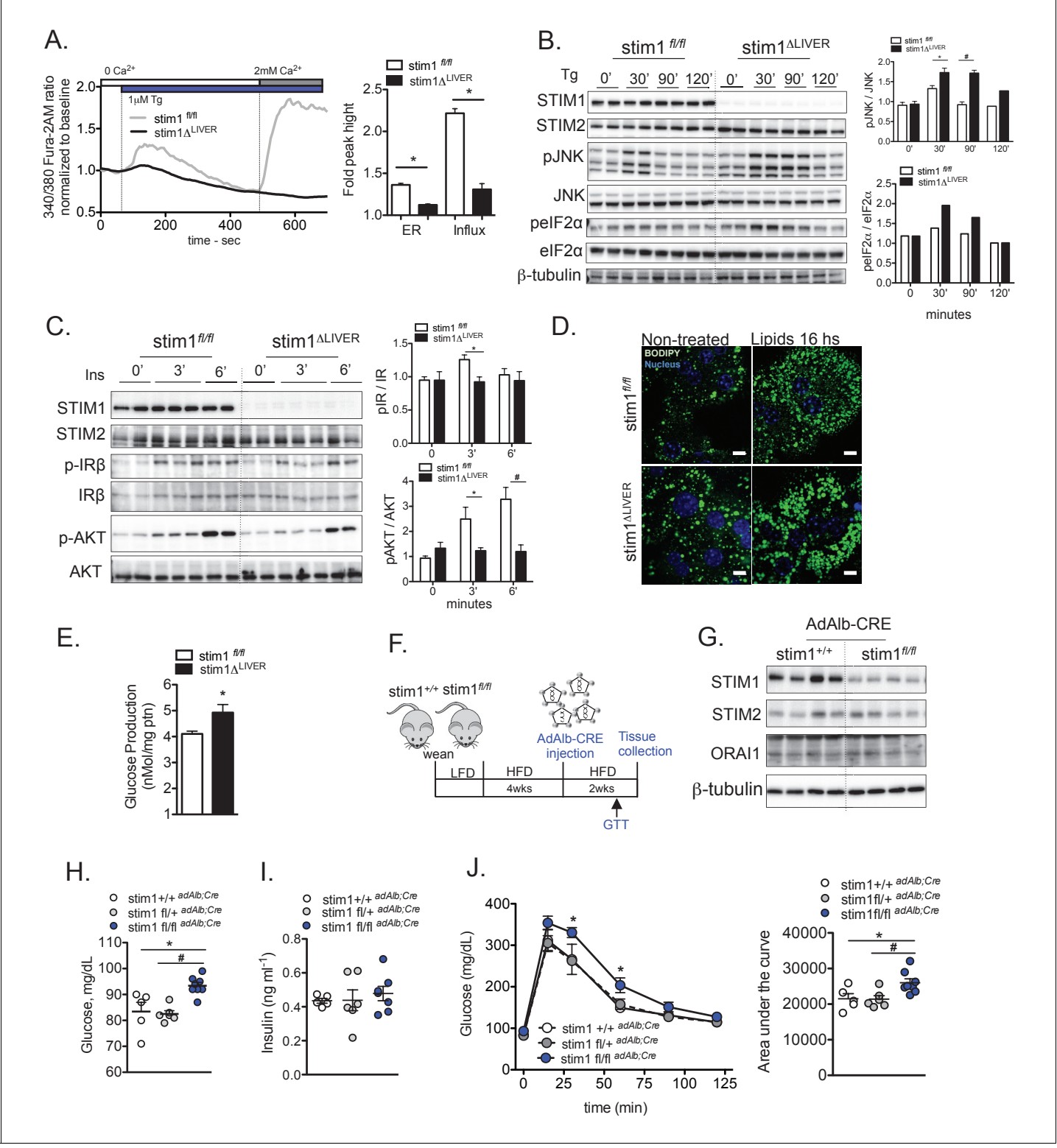

**Figure 3.** STIM1 deficiency leads to increased cellular stress, insulin resistance and lipid droplet accumulation. (**A**) Left panel: representative Fura-2AM-based cytosolic $Ca^{2+}$ measurements in primary hepatocytes isolated from stim1$^{fl/fl}$ and stim1$\Delta^{LIVER}$ mice. Right panel: quantification of Tg-induced $Ca^{2+}$ release (reflecting ER $Ca^{2+}$ content) and SOCE. Error bars denote s.e.m. n = 98 WT and n = 116 stim1$\Delta^{LIVER}$ for Tg response and n = 69 stim1$^{fl/fl}$ and n = 101 stim1$\Delta^{LIVER}$ for SOCE. Cells were pooled across four independent experiments. *p<0.0001 (**B**) Immunoblot analysis and quantification of the expression levels of stress markers in primary hepatocytes isolated from stim1$^{fl/fl}$ and stim1$\Delta^{LIVER}$ mice. n = 6 samples in each group, pooled across

*Figure 3 continued on next page*

Figure 3 continued

three independent experiments for pJNK and n = 2 for peIF2α, *p=0.0138 #p<0.0001. (C) Immunoblot analysis and quantification of insulin signaling before and after 3 nM insulin (Ins) treatment for the indicated time points in primary hepatocytes isolated from stim1$^{fl/fl}$ and stim1Δ$^{LIVER}$ mice. n = 4–5 samples in each group, pooled across two independent experiments for pIR. *p=0.011 and pAKT *p=0.03 and #p=0.0088 (D) Primary hepatocytes isolated from stim1$^{fl/fl}$ and stim1Δ$^{LIVER}$ animals treated or not with a mixture containing 1 mM Oleic Acid and 40 µM palmitic acid for 16 hr. Lipid droplets were stained in green with BODIPY and the nucleus in blue with DAPI. Scale: 10 µm. (E) Glucose output derived from primary hepatocytes isolated from stim1$^{fl/fl}$ and stim1Δ$^{LIVER}$ mice kept on a HFD, n = 4 stim1$^{fl/fl}$ and n = 4 stim1Δ$^{LIVER}$*p=0.046 (F) Schematic representation of the protocol for adenovirus-mediated transient hepatocyte knockdown of STIM1 (G) Immunoblot analysis of the indicated proteins in total liver lysates derived from stim1$^{+/+}$ n = 4 and n = 4 stim1$^{fl/fl}$ mice, both groups expressing liver-specific Cre recombinase (ad Alb;Cre) adenovirus. (H and I) 16 hr fasting blood glucose levels (H) and insulin (I) in animals of the indicated genotypes. *p=0.0094 stim1$^{+/+}$ versus stim1$^{fl/fl}$ and #p<0.0001 stim1$^{fl/+}$ versus stim1$^{fl/fl}$ (J) Glucose tolerance test (GTT) in stim1$^{+/+}$, stim1$^{fl/+}$ and stim1$^{fl/fl}$ animals expressing adenoviral Alb;Cre recombinase *p<0.04; Right panel: quantification of area under the curve from GTT. n = 5 stim1$^{+/+}$, n = 6 stim1$^{fl/+}$, n = 8 stim1$^{fl/fl}$ animals, *p=0.03 stim1$^{+/+}$, versus stim1$^{fl/fl}$ and #p=0.01 stim1$^{fl/+}$ versus stim1$^{fl/fl}$, representative of 2 independent experiments. For all graphs, error bars denote s.e.m.

DOI: https://doi.org/10.7554/eLife.29968.013

The following source data and figure supplements are available for figure 3:

**Source data 1.** Source data for *Figure 3*.
DOI: https://doi.org/10.7554/eLife.29968.014

**Figure supplement 1.** STIM1 downregulation increases cellular stress and impairs insulin signaling in Hepa 1-6 cells.
DOI: https://doi.org/10.7554/eLife.29968.015

**Figure supplement 1—source data 1.** Source data for *Figure 3—figure supplement 1*.
DOI: https://doi.org/10.7554/eLife.29968.016

**Figure supplement 2.** Effect of liver-specific deletion of STIM1 on metabolic parameters in vivo.
DOI: https://doi.org/10.7554/eLife.29968.017

**Figure supplement 2—source data 1.** Source data for *Figure 3—figure supplement 2*.
DOI: https://doi.org/10.7554/eLife.29968.018

indicate that increased modification of STIM1 by OglcNAc could, at least in part, underlie defective STIM1 translocation and reduced hepatic SOCE in the context of obesity.

We then started to investigate the impact of defective STIM1-mediated SOCE on ER homeostasis, cellular stress responses and metabolic regulation in STIM1-deficient hepatocyte cell models. These included primary hepatocytes derived from mice with genetic STIM1 deficiency specifically in the liver (stim1$^{fl/fl}$ Alb;Cre, identified here as stim1Δ$^{LIVER}$) and Hepa1-6 cells stably expressing shRNAs targeting stim1 (*Figure 3—figure supplement 1A*) or stim2 (*Figure 3—figure supplement 1B*) (identified here as shSTIM1 and shSTIM2 respectively). Primary hepatocytes derived from stim1-Δ$^{LIVER}$ mice displayed lower ER Ca$^{2+}$ content and absence of Tg-triggered SOCE compared to controls (stim1$^{fl/fl}$) (*Figure 3A*). Similarly, Hepa 1–6 cells in which STIM1 expression was down regulated showed markedly blunted SOCE (*Figure 3—figure supplement 1C*). Imbalances in ER Ca$^{2+}$ content trigger cellular stress responses through various pathways (*Arruda and Hotamisligil, 2015*; *Fu et al., 2012*; *Ozcan and Tabas, 2016*). In order to examine these responses in our cellular models of STIM deficiency, we measured phosphorylation of JNK as a benchmark measure of cellular stress and inflammatory activation, and phosphorylation of eIF2α as a marker of UPR activation. As shown in *Figure 3B*, in stim1$^{fl/fl}$ control cells, treatment with Tg induced phosphorylation of JNK and eIF2α at 30 min followed by a decrease in phosphorylation levels back to baseline after 90 min. However, in STIM1-deficient cells, the phosphorylation of JNK and eIF2α was enhanced at baseline and these cells displayed a stronger and more persistent response to Tg, with stress markers only returning to basal levels at 120 min. A similar profile was observed in Hepa1-6 cells with shRNA-mediated suppression of STIM1 (*Figure 3—figure supplement 1D*). Interestingly, suppression of STIM2 alone did not alter the cellular response to Tg (*Figure 3—figure supplement 1E*). Altogether, these data demonstrate that absence of core components of SOCE leads to elevated and prolonged stress responses in hepatocytes.

Increased cellular stress and inflammation are associated with impaired insulin action and defective glucose and lipid metabolism (*Fu et al., 2012*; *Hirosumi et al., 2002*; *Hotamisligil, 2017*). Accordingly, we found that primary hepatocytes derived from stim1Δ$^{LIVER}$ mice displayed impaired phosphorylation of IRβ and AKT in response to insulin relative to cells from stim1$^{fl/fl}$ littermates (*Figure 3C*). Likewise, gene silencing of stim1 in Hepa1-6 cells resulted in decreased insulin signaling

(*Figure 3—figure supplement 1F*). Additionally, primary hepatocytes isolated from stim1Δ$^{LIVER}$ mice maintained on HFD showed higher levels of glucose production stimulated by the gluconeogenesis substrates glycerol, pyruvate and glutamine compared to stim1$^{fl/fl}$ derived hepatocytes (*Figure 3E*). In agreement with our finding that STIM2 suppression does not amplify stress responses, we found that STIM2-deficient cells retained normal insulin responsiveness (*Figure 3—figure supplement 1G*).

Recently, it has been shown that SOCE may regulate lipid metabolism (*Maus et al., 2017*; *Wilson et al., 2015*) and animals with inducible STIM1/2 whole body deficiency exhibit increased lipid accumulation in multiple tissues such as skeletal muscle, heart and liver, as a consequence of impaired lipolysis and fatty acid oxidation (*Maus et al., 2017*). We observed that primary hepatocytes isolated from stim1Δ$^{LIVER}$ mice showed modestly increased accumulation of lipid droplets at baseline and after incubation with 1 mM oleic acid and 40 µM of palmitic acid (*Figure 3D*). These results suggest that in addition to higher stress levels, insulin resistance and increased glucose production, SOCE deficiency may result in abnormal lipid metabolism.

We next assessed the systemic metabolic impact of STIM1 deficiency in hepatocytes in vivo in the stim1Δ$^{LIVER}$ mice described above. As shown in *Figure 3—figure supplement 2A*, liver lysates from stim1Δ$^{LIVER}$ mice displayed marked reduction in STIM1 and a mild increase in STIM2 protein. On a chow diet, stim1Δ$^{LIVER}$ mice gained weight equivalently to stim1$^{fl/fl}$ mice (*Figure 3—figure supplement 2B*) and did not exhibit alterations in glucose tolerance and triglyceride content (Tg) (data not shown). In the HFD-fed mice, the weight gain was similar between genotypes (*Figure 3—figure supplement 2B*). Interestingly, at 6 weeks on a HFD, we observed a mild, but significant, glucose intolerance in the stim1Δ$^{LIVER}$ animals compared with the stim1$^{fl/fl}$ controls (*Figure 3—figure supplement 2C*). Additionally, at this time point, insulin sensitivity was impaired (*Figure 3—figure supplement 2D*). No significant differences were observed in Tg content between the two genotypes, although a tendency to higher Tg levels was detected in stim1Δ$^{LIVER}$ mice (*Figure 3—figure supplement 2E*). In long term HFD (20 weeks), the differences in glucose intolerance, insulin signaling and Tg content between stim1$^{fl/fl}$ and stim1Δ$^{LIVER}$ groups were very mild and did not reach statistical significance (*Figure 3—figure supplement 2F,G and H*).

Interestingly, although we haven't observed overall changes in lipid accumulation in the liver specific STIM1 deficient animals, we have noticed that these animals showed higher content of microvesicular steatosis compared with controls where the majority of the lipid droplets are presented as large droplets (*Figure 3—figure supplement 2I*).

In light of the mild in vivo metabolic phenotype of constitutive hepatocyte-specific STIM1 deletion during development, we considered that STIM2 upregulation (*Figure 3—figure supplement 2*) or alternative systems could compensate for the lack of STIM1 all throughout the embryonic life. Therefore, we examined the effect of acute deletion of STIM1 in hepatocytes, using adenovirus mediated gene delivery to express albumin-Cre recombinase (*Alb;Cre*) in adult stim1$^{+/+}$ and stim1$^{fl/fl}$ mice. After weaning, littermate animals were subjected to HFD for 4 weeks (for a short-term stress induction) (*Figure 3F*). Adenoviral delivery of *Alb;Cre* recombinase resulted in ~60% deletion of hepatic STIM1 (*Figure 3G*). Importantly, there was no compensatory upregulation of STIM2 or Orai1 in this setting. Indeed, one week after adenovirus administration, the acute deletion of STIM1 resulted in significantly higher fasting glucose levels (*Figure 3H*) without any difference in insulin levels (*Figure 3I*) and body weight (data not shown). Interestingly, following STIM1 deletion, mice also exhibited significantly impaired glucose tolerance (*Figure 3J*). In summary, acute SOCE dysfunction in the liver results in amplified cellular stress and glucose intolerance in mice fed a HFD.

Based on these observations, and the marked SOCE defect associated with obesity we hypothesized that recovery of SOCE function in the livers of obese mice could improve metabolic homeostasis in obesity. To test this hypothesis, we used a hepatocyte-specific adenovirus system to exogenously express either GFP (control) or STIM1-YFP in the *Lep$^{ob/ob}$* primary hepatocytes and mice (*Figure 4A*). The functionality of the STIM1-YFP fusion protein has been verified previously (*Liou et al., 2005*) and confirmed by us (*Video 1*).

Using this approach, we asked whether exogenous expression of STIM1-YFP would be sufficient to overcome the SOCE defects in primary hepatocytes from obese animals. As shown in *Figure 4B*, in control *Lep$^{ob/ob}$* cells expressing GFP, STIM1 formed puncta at baseline and Tg-induced translocation of the protein was impaired, as we previously observed (*Figure 1D*). Delivery of STIM1-YFP lead to a significant increase in the protein level in liver cells. Interestingly, in non treated cells, we observed that some degree of the STIM1 protein was already localized in areas of close contact with

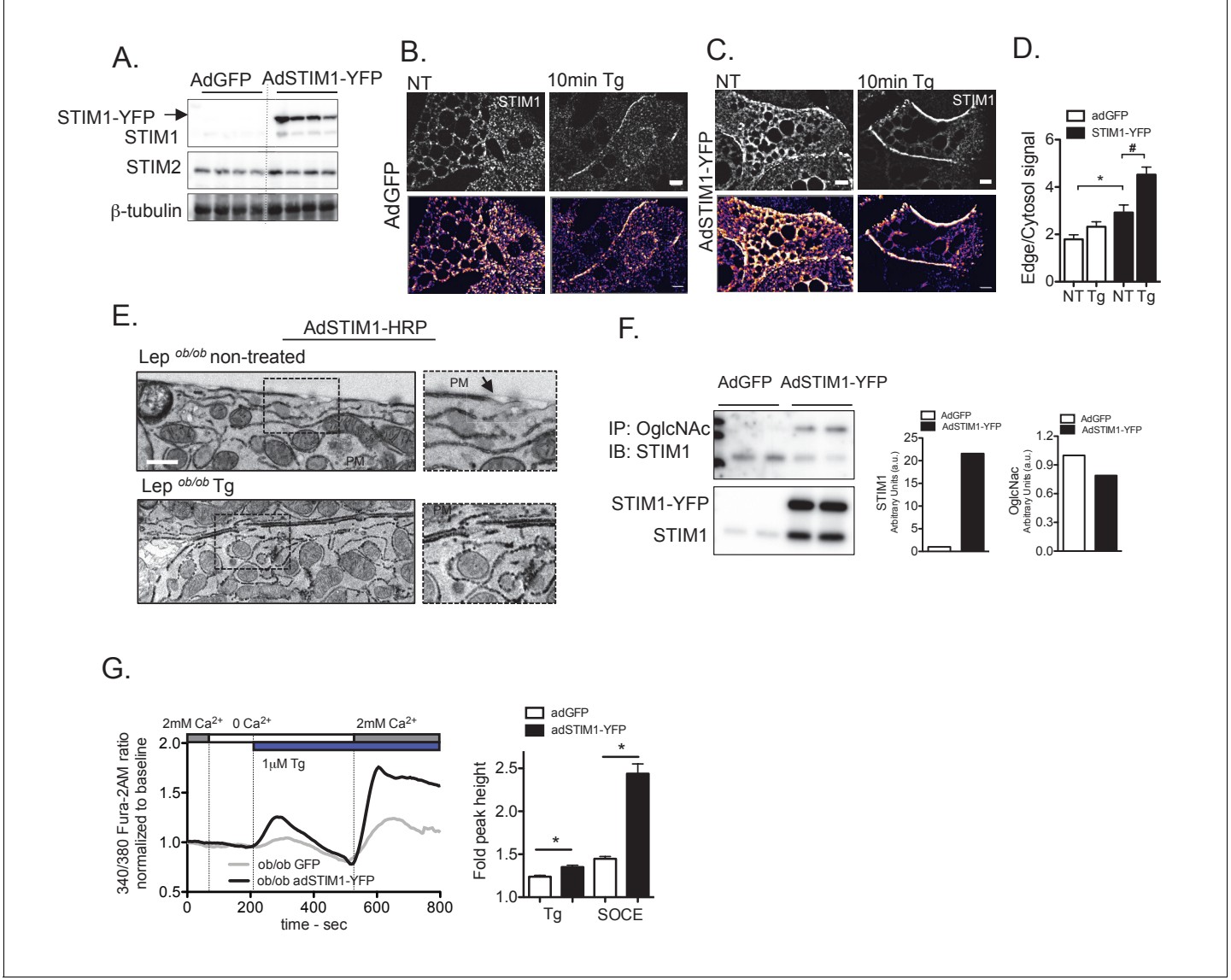

**Figure 4.** Overexpression of STIM1 promotes STIM1 translocation to ER/PM junctions and improves SOCE in primary hepatocytes from obese animals. (**A**) Immunoblot analysis of STIM1-YFP expression levels in total lysates from livers isolated from *Lep^ob/ob^* mice transduced with adenovirus (Ad) expressing GFP or STIM1-YFP. (**B–C**) Confocal images of immunofluorescence staining for endogenous STIM1 in primary hepatocytes from *Lep^ob/ob^* mice expressing adGFP (**B**) or adSTIM1-YFP (**C**) treated with DMSO (vehicle) and 1 µM Tg. NT refers to 'not Tg treated'. (**D**) Quantification of STIM1 translocation by calculating the ratio between the mean STIM1 pixel intensity at a selected area of the edge of the cell relative to the same measurement performed in the cytosol. n = 26 adGFP NT and Tg treated and n = 16 adSTIM1-YFP NT and n = 19 Tg treated *p=0.0023, #p=0.0015. Scale: 10 µm. (**E**) Transmission Electron Micrographs of primary hepatocytes isolated from *Lep^ob/ob^* animals expressing STIM1-HRP, treated with DMSO or treated 1 µM Tg for 10 min. Scale: 500 nm. (**F**) Immunoblot analysis and densitometric quantification of OglcNac and STIM1 expression in primary hepatocytes from *Lep^ob/ob^* animals infected with AdGFP and AdSTIM1-YFP adenovirus for 24 hr. Quantification reflects the sum of the endogenous and exogenous STIM1 (**G**) Left: Representative Fura-2AM-based cytosolic $Ca^{2+}$ measurements in primary hepatocytes isolated from *Lep^ob/ob^* animals expressing adGFP or adSTIM1-YFP. Right: Quantification of Tg-induced $Ca^{2+}$ release (reflecting ER $Ca^{2+}$ content) and SOCE measurements . n = 126 adGFP and n = 83 adSTIM1-YFP cells for Tg response and n = 127 adGFP and n = 62 adSTIM1-YFP cells for SOCE, representative of 5 independent experiments. *p<0.0001. For all graphs. error bars denote s.e.m.

DOI: https://doi.org/10.7554/eLife.29968.019

The following source data and figure supplement are available for figure 4:

**Source data 1.** Source data for *Figure 4*.
DOI: https://doi.org/10.7554/eLife.29968.020

**Figure supplement 1.** Effect of STIM1-YFP overexpression in ER morphology and apposition to the plasma membrane.
DOI: https://doi.org/10.7554/eLife.29968.021

the plasma membrane even prior to stimulation (*Figure 4C*). Additionally, exogenously expressed STIM1 was able to translocate towards the plasma membrane after Tg treatment (*Figure 4C and D*). To examine STIM1 translocation in *Lep^{ob/ob}* hepatocytes with higher resolution, we also performed TEM in cells expressing exogenous STIM1 fused with HRP. STIM1-HRP-expressing cells were fixed and treated with diaminobenzidine (DAB) in the presence of $H_2O_2$. HRP catalyzes the polymerization and deposition of DAB, which recruits electron-dense osmium, providing contrast and revealing the localization of STIM1. As shown in *Figure 4E*, exogenous delivery of STIM1-HRP resulted in rescue of translocation in cells from obese animals in a manner similar to that observed in lean, wild type cells. Taken together, these data indicate that exogenously increasing the amount of STIM1 protein is sufficient to partially overcome the STIM1 translocation defect in cells from obese mice.

The ability of STIM1-YFP overexpression to overcome, at least in part, the translocation defect of endogenous STIM1 in *Lep^{ob/ob}* cells suggests that at least some proportion of STIM1 in this setting may escape the O-GlycNac modification. To examine the degree of O-GlcNacylation of overexpressed STIM1-YFP in *Lep^{ob/ob}* cells, we used an O-GlcNac specific antibody to immunoprecipitate all proteins modified by O-GlycNacylation in GFP or STIM1 YFP overexpressing cells and examined STIM1 protein by immunoblotting. As shown in *Figure 4F*, STIM1-YFP overexpression resulted in an ~20 fold increase in the expression of STIM1 protein. However, the amount of O-GlcNac modified STIM1 did not increase proportionally, indicating that in fact a significant amount of the overexpressed STIM1 escapes this post translation modification.

Additionally, it has been previously shown that overexpression of STIM1 in HeLa cells leads to morphological changes in the ER with an increased cortical ER (*Orci et al., 2009*). In agreement with this finding, electron micrographs of primary hepatocytes derived from *Lep^{ob/ob}* mice show that STIM1-YFP expression led to a remodeling of ER with abundant ER stacks apposed to the PM, although the degree of ER remodeling varied from cell to cell, likely due to variable STIM1 expression levels (*Figure 4—figure supplement 1A*). This result suggests that, in addition to having more STIM1 that translocates to PM, overexpression of STIM1-YFP also remodels the ER, favoring the proximity of STIM1 to Orai at the plasma membrane.

After verifying that overexpression of STIM1-YFP was able to rescue STIM1 translocation defects in obese cells, we examined whether this actually resulted in improved SOCE. As shown in *Figure 4G* overexpression of STIM1-YFP in *Lep^{ob/ob}* cells resulted in increased ER $Ca^{2+}$ and SOCE compared to control cells. Thus, overexpression of STIM1 was a successful intervention to overcome the STIM translocation and SOCE defects, resulting in improved ER calcium handling.

Next, we asked whether the effects of the overexpression of STIM1-YFP in *Lep^{ob/ob}* primary hepatocytes would impact overall metabolism in vivo, and evaluated the effect of liver-specific STIM1 expression on systemic glucose metabolism in the *Lep^{ob/ob}* mice following the protocol displayed in *Figure 5A*. Introduction of STIM1 by adenovirus gene delivery led to increased STIM1 protein levels in the liver. Interestingly, increased STIM1 expression was also accompanied by increased mRNA expression of Orai1 and with an increase in SERCA (Atp2a2) mRNA and protein levels (*Figure 5B and C*). This is in accordance with previous work showing that the modulation of expression STIM1 affects the expression of other $Ca^{2+}$ channels or pumps, as SERCA, likely due to alterations in cytosolic $Ca^{2+}$ levels (*Abell et al., 2011*). Remarkably, liver-specific overexpression of STIM1 led to a significant improvement in glucose tolerance relative to control (adGFP) animals as evaluated by a glucose tolerance test (*Figure 5D*). Improved glucose tolerance resulting from overexpression of STIM1 was associated with enhanced in vivo insulin signaling evaluated by direct insulin injection into the livers that resulted in increased AKT

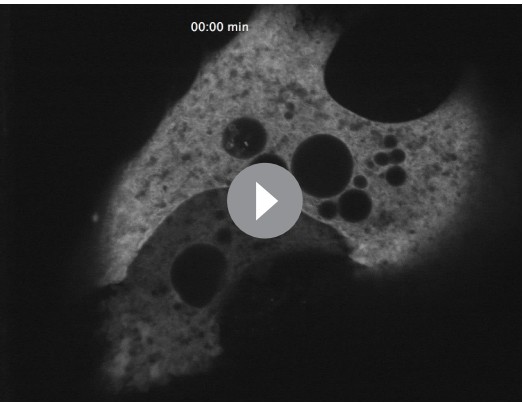

**Video 1.** STIM1-YFP translocation upon thapsigargin (Tg) treatment. Primary Hepatocytes were infected with STIM1-YFP, treated with 1 uM Tg and YFP fluoresce was recorded over the period of time indicated in the movie.
DOI: https://doi.org/10.7554/eLife.29968.022

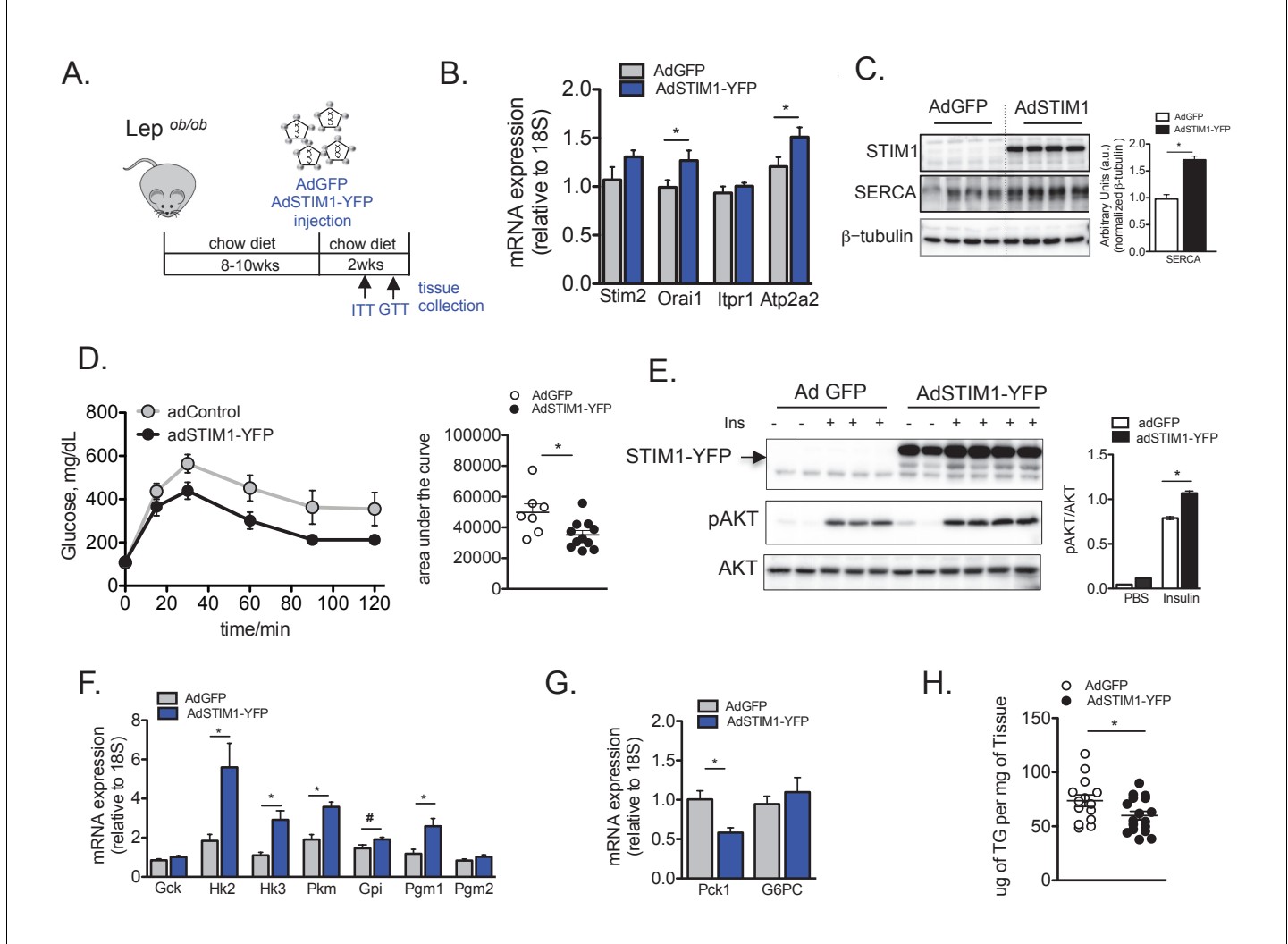

**Figure 5.** Overexpression of STIM1 leads to improved insulin signaling, glucose tolerance and lipid accumulation in obese animals. (**A**) Schematic representation of the protocol for adenovirus mediated STIM1- YFP expression in *Lep*[ob/ob] mice (**B**) mRNA expression levels of indicated genes evaluated by qPCR, normalized to 18S, n = 9 for WT and n = 10 for STIM1-YFP pooled from two independent experiments, *p=0.04 (**C**) Left panel: Immunoblot analysis of indicated proteins in liver total lysates from *Lep*[ob/ob] mice expressing GFP or STIM1-YFP. Right panel: quantifications of the blots. n = 3 adGFP and n = 4 adSTIM1-YFP, *p=0.001 (**D**) Glucose tolerance tests in *Lep*[ob/ob] mice after adGFP (control) and adSTIM1-YFP expression with quantifications of area under the curve. n = 7 adGFP and n = 11 adSTIM1-YFP animals, representative of 3 independent experiments.*p=0.022 (**E**) Markers of in vivo insulin signaling evaluated by immunoblot analysis of total liver lysates from animals injected with insulin (0.75 U/kg) through the portal vein. Tissues were collected 3 min after injection. Right panel: phospho-protein level quantification normalized to total protein levels. n = 3 for adGFP and n = 4 for adSTIM1-YFP, *p=0.0003. (**F and G**) mRNA expression levels of indicated genes evaluated by qPCR, normalized to 18S, n = 9 for both WT and STIM1-YFP pooled from two independent experiments. *p<0.01 # p<0.04. (**H**) Liver triglyceride content in *Lep*[ob/ob] animals expressing adGFP (control) and adSTIM1-YFP pooled from three independent cohorts. n = 14 adGFP and 17 adSTIM1-YFP, *p=0.042. For all graphs, error bars denote s.e.m.

DOI: https://doi.org/10.7554/eLife.29968.023

The following source data and figure supplements are available for figure 5:

**Source data 1.** Source data for *Figure 5*.

DOI: https://doi.org/10.7554/eLife.29968.024

**Figure supplement 1.** Effect of STIM1-YFP overexpression in mRNA expression of genes involved in lipolysis and lipogenesis.

DOI: https://doi.org/10.7554/eLife.29968.025

**Figure supplement 1—source data 1.** Source data for *Figure 5—figure supplement 1*.

DOI: https://doi.org/10.7554/eLife.29968.026

phosphorylation (*Figure 5E*). Additionally, expression of genes involved in glycolysis were increased in livers overexpressing STIM1, accompanied by decreased levels of PCK1 (PEPCK), a rate limiting enzyme for gluconeogenesis (*Figure 5F and G*).

Given the reported role of STIM1 in lipid metabolism, we also evaluated if overexpression of STIM1 impacted the excessive lipid accumulation in the livers of *Lep^ob/ob* animals. As shown in *Figure 5H*, exogenous expression of STIM1 led to a reduction in total Tg content in the liver tissue of *Lep^ob/ob* mice compared to controls expressing GFP. Accordingly, STIM1 overexpression led to increase mRNA expression of genes involved in fatty acid oxidation, such as CPT1, while no change was observed in genes involved in lipogenesis (*Figure 5—figure supplement 1A and B*). Based on these results we conclude that SOCE is critical for glucose and lipid metabolism and that increasing hepatic SOCE through overexpression of STIM1 is able to revert, at least in part, the deleterious impact of obesity on systemic glucose homeostasis.

## Discussion

In the last several years, a growing number of studies have laid the foundation for the concept that disrupted intracellular $Ca^{2+}$ homeostasis in metabolic tissues is a key component of ER dysfunction and metabolic deterioration (*Arruda and Hotamisligil, 2015*; *Ozcan and Tabas, 2016*). The abnormal $Ca^{2+}$ handling in the ER of hepatocytes in this setting is multi-faceted, involving both impairment of SERCA activity (*Fu et al., 2011*) as well as increased activity of IP3R (*Arruda et al., 2014*; *Feriod et al., 2017*; *Wang et al., 2012*). This impacts not only $Ca^{2+}$ levels in the ER and protein folding, but also mitochondrial oxidation and ROS production as well as the activation of cytosolic stress signaling pathways. Supporting the critical role of this biology, human genetic studies have shown that SNPs in SERCA (*Varadi et al., 1999*) and IP3R (*Shungin et al., 2015*) are associated with metabolic diseases such as obesity and diabetes. Here, our work reveals that another key mechanism supporting ER $Ca^{2+}$ homeostasis, STIM-mediated SOCE, is also defective in obesity with important implications for the metabolic dysfunction in hepatocytes. Strikingly, we demonstrate that STIM1 gain-of-function improves cellular stress responses and glucose homeostasis in obese mice, underscoring the therapeutic potential of targeting $Ca^{2+}$ homeostasis through STIM1 for metabolic disease treatment.

Our interest in this biology was heightened by our initial observation that in hepatocytes isolated from obese (*Lep^ob/ob*) mice, STIM1 was present in an aberrant punctate pattern along the ER membrane. One possible explanation was that *Lep^ob/ob* hepatocytes display increased SOCE even in baseline conditions in order to respond to low ER $Ca^{2+}$ levels induced by dysfunction of SERCA and IP3R1. However, our detailed examination revealed that in cells of obese mice, STIM1 puncta were not localized in areas of close proximity of the PM, but rather were distributed all throughout the cell away from the PM. Additionally, when we depleted ER $Ca^{2+}$ stores, STIM1 was not able to fully translocate to the PM, resulting in significantly reduced SOCE. Hence, metabolic stress impairs this critical component of ER $Ca^{2+}$ homeostasis in obese liver tissue.

One question arising from these observations is the nature of the mechanism underlying STIM1 dysfunction in obesity. STIM activation and translocation is modulated by a complex array of factors that include post-translational modification of the protein itself, protein-protein interactions (for review *Lopez et al., 2016*), the interaction of STIM1 with lipids of the ER membrane and the ability of the ER itself to remodel and form ER/PM junctions (*Derler et al., 2016*; *Lopez et al., 2016*; *Prakriya and Lewis, 2015*). Here we found that in hepatocytes derived from *Lep^ob/ob* mice, STIM1 displays significantly increased levels of O-GlcNAcylation, a post-translation modification that causes impaired STIM1 trafficking and function (*Zhu-Mauldin et al., 2012*). Indeed, overexpression of OGT in hepatocytes was sufficient to induce abnormal STIM1 puncta formation and impaired activation of SOCE while down-regulation of OGT in primary hepatocytes from obese mice partially rescued the STIM1 translocation defect. It will be interesting to explore the detailed location and impact of these modifications in future studies. In silico analysis, published literature and data in dbOGAP, suggest potential target sites coinciding with the domains involved in STIM1/Orai1 interactions with Orai1 in the CAD/SOAR domain or in the polybasic motif located at the C terminal of the protein that may be critical in oligomerization and puncta formation (*Liou et al., 2007*). Regardless, protein modification with O-GlcNAc is a reflection of the nutritional status, as it integrates glucose, amino acid, fatty acid and nucleotide flux through the hexosamine biosynthetic pathway (HBP) to produce *N*-acetyl-

glucosamine (UDP-GlcNAc), the obligatory substrate for OGT. Hence, conditions of nutrient and substrate excess, including obesity, lead to increased cellular O-GlcNAcylation levels (*Vosseller et al., 2002*; *Yang and Qian, 2017*; *Dentin et al., 2008*; *Yang et al., 2008*). Taken together, our findings demonstrate that O-GlcNAc modification of STIM1 alters organelle $Ca^{2+}$ regulation with consequences for ER function and systemic metabolism and provide an important mechanistic insight into regulation of this critical metabolic integration point through organelle homeostasis in obesity.

Another important finding reported here is that impaired SOCE in hepatocytes by STIM1 deficiency was sufficient to induce markers of ER stress and inflammation, to impair of insulin action, increase glucose production and induce excessive lipid droplet accumulation. Recently, it has been reported that cells from patients with loss-of-function mutations in STIM1 or ORAI accumulate lipid droplets as a consequence of their inability to increase cAMP, mobilize fatty acids from lipid droplets, activate lipolysis, and oxidize fatty acids (*Maus et al., 2017*). Interestingly, tamoxifen-induced whole body STIM1/2 deficiency leads to higher lipid content in the heart, muscle and liver (*Maus et al., 2017*). Here, although we detected increased accumulation of lipid droplets in hepatocytes lacking STIM1 in vitro, these phenomena were not clearly observed in vivo. Hepatocyte-specific deletion of STIM1 led to a mild metabolic phenotype in vivo suggesting that the activation of compensatory systems occurs when STIM1 is deleted during development. Indeed, STIM2 expression levels were slightly increased in STIM1 deficient liver. Furthermore, in lean animals, the rest of the ER calcium handling systems are intact, which may allow maintenance of the normal equilibrium. In contrast, acute down-regulation of STIM1 showed a more marked effect on glucose homeostasis.

Finally, we show here that exogenous expression of STIM1 in primary hepatocytes from obese mice is sufficient to overcome SOCE defects and partially correct ER $Ca^{2+}$ levels and STIM1 re-localization upon $Ca^{2+}$ store depletion. Notably, we showed that a significant part of the exogenously expressed STIM1 is able to escape the O-GlcNAc modification, probably by a mass effect, and responds properly to ER $Ca^{2+}$ depletion. Additionally, overexpression of STIM1 in hepatocytes from $Lep^{ob/ob}$ animals led to remodeling of the ER with an increase in the amount of cortical ER possibly facilitating STIM1 translocation and Orai1 coupling. Indeed, it is known that STIM1 regulates the structure of the ER through its role in Tip Attachment Complex movement (*Grigoriev et al., 2008*; *Westrate et al., 2015*), coordinating the growth and shrinkage of the ER tubule. Importantly, we show that upregulation of SOCE through STIM1 replenishment in the liver of obese mice results in significant metabolic benefit including improved insulin signaling and glucose tolerance and decreased lipid content likely as a result of enhanced fatty acid oxidation. When STIM1 is exogenously supplied to the liver tissue in obese mice, we also observed an increase in glycolytic gene expression. This is in agreement with recent report showing that in T cells SOCE regulates the expression of glycolytic genes through the modulation of the transcription factor NFAT (*Vaeth et al., 2017*). Overexpression of STIM1 also led to increased SERCA levels, which is necessary for the delivery of $Ca^{2+}$ into the ER. In fact, it is proposed and validated experimentally and computationally that an adaptive feedback loop between components of $Ca^{2+}$ machinery exist to keep cytosolic $Ca^{2+}$ levels under control (*Abell et al., 2011*). Increased SERCA expression in the context of STIM1 overexpression may indeed be a critical component of the metabolic benefit resulting from STIM1 expression in the liver in obese mice. It is remarkable that the individual manipulation of multiple proteins involved in the maintenance of ER $Ca^{2+}$ levels- SERCA, IP3R, and STIM1- all result in a similar phenotype. These findings underscore the importance of ER $Ca^{2+}$ homeostasis and ER function to metabolic health, and indicate the strong potential for targeting these pathways to combat metabolic disease.

## Materials and methods

### General animal care and study design

All in vivo studies are approved by the Harvard Medical Area Standing Committee on Animals. Unless stated otherwise, mice were maintained from 4 to 20 weeks on a 12-hour-light/12 hr-dark cycle in the Harvard T.H. Chan School of Public Health pathogen-free barrier facility with free access to water and to a standard laboratory chow diet (PicoLab Mouse Diet 20 #5058, LabDiet). No specific power analysis was used to estimate sample size. The sample size and number of replicates for

this study were chosen based on previous experiments performed in our lab and others (*Arruda et al., 2014*; *Fu et al., 2011*).

## Animal models of obesity

We used two mouse models of obesity, the leptin-deficient $Lep^{ob/ob}$ mouse, and HFD-induced obesity. For the former, wild-type mice in the C57BL/6J genetic background (Stock no. 000664) and $Lep^{ob/ob}$ mice (Stock no. 000632) were purchased from Jackson Laboratories at 6–7 weeks of age and used for experimentation between 8–12 weeks of age. For the latter, male C57BL/6J mice were purchased from Jackson Laboratories and placed on HFD (D12492: 60% kcal% fat; Research Diets) for up to 20 weeks. Control mice of the same age were fed with a low fat diet (PicoLab Mouse Diet 20 #5053, LabDiet). In animal experiments, all measurements were included in the analysis unless they fell more than two standard deviations from the mean.

## STIM1 hepatocyte-specific deficient mice

Mice carrying floxed alleles for stim1 (referred here as $stim1^{fl/fl}$) on a C57BL/6J background were kindly provided by Dr. Anjana Rao, La Jolla Institute for Allergy and Immunology. To generate hepatocyte-specific STIM deficient mice, $stim1^{fl/fl}$ mice were bred to C57BL/6J mice expressing CRE recombinase under the control of the albumin promoter (*Alb;Cre*). *Alb;Cre* -mediated recombination of floxed stim1 alleles was detected in genomic DNA by PCR. Age-matched littermates were used for the study of adult mice. For HFD studies, pups were placed on low-fat control diet (5053) for 1 week at weaning, followed by up to 20 weeks on HFD. The control chow group remained on low-fat diet. For the adenovirus-mediated stim1 down-regulation study, $stim1^{fl/+}$ (het) animals were crossed to each other to generate a group containing $stim1^{+/+}$, $stim1^{fl/+}$ and $stim1^{fl/fl}$ homozygous mice. Pups of the three genotypes were weaned and fed a low-fat diet (5053) for 1 week followed by 4 weeks of HFD. The animals were transferred to a BL2 facility where they received adenovirus-expressing *Alb;Cre*-recombinase ($1 \times 10^9$ IFU/mouse) intravenously. Metabolic studies were performed between day 7–12 after infection. The animals were sacrificed after 14 days of infection for ex vivo experiments.

## Adenovirus-mediated overexpression of STIM1

For exogenous STIM1 expression, purified, de-salted adenovirus (serotype 5, Ad5) expressing STIM1-YFP was purchased from Vector BioLabs with the agreement of Dr. Alexei Tepikin from University of Liverpool. The adenovirus was administered to 8–10 week-old $Lep^{ob/ob}$ mice intravenously, at a titer of $1 \times 10^9$ IFU/mouse. Metabolic studies were performed between day 7–12 after infection. The animals were sacrificed after 14 days of infection for ex vivo experiments.

## Glucose tolerance tests

Animals were subjected to an intraperitoneal (i.p.) glucose injection (lean: 1.5 g kg$^{-1}$, obese: 0.5–1.0 g kg$^{-1}$) after overnight fasting, and blood glucose levels were measured throughout the first 120 min of the metabolic response.

## Liver triglyceride measurements

Liver tissues (approximately 100 mg) were homogenized in 1.2 mL of water. Next, 100 μL of the homogenate was transferred to a 1.5 mL tube and 125 μL of chloroform and 250 μL of Methanol were added. Samples were vortexed briefly and incubated for 5 min. Additional 125 μL of chloroform was added. Next, 125 μL of water was added and samples were vortexed and centrifuged at 3000 rpm for 20 min at 4°C. Around 150 μL of the lower phase was collected in a 1.5 mL tube and dried (evaporated) in a heated vacuum oven. Thereafter, lipids were re-suspended in 300 uL of ethanol. Triglyceride in this solution was measured by a Randox Tg Kit (cat number TR213) from Randox Laboratories.

## Primary hepatocyte isolation

Animals were anesthetized using 2 mg/ml xylazine combined with 2 mg/ml ketamine in PBS and the livers were perfused with 50 mL of buffer I (content described below) through the portal vein with an osmotic pump set to the speed of ~4 mL/min until the liver turned pale. The speed was gradually

**Table 1.** List of primers used for qPCR measurements and sequences of the shRNA used in this study.

**List of SYBR green primers for real-time PCR**

| Gene name | Forward primer | Reverse primer |
| --- | --- | --- |
| stim1 | TGAAGAGTCTACCGAAGCAGA | AGGTGCTATGTTTCACTGTTGG |
| stim1 | ACAGTGAAACATAGCACCTTCC | TCAGTACAGTCCCTGTCATGG |
| stim2 | CGAAGTGGACGAGAGTGATGA | GGAGTGTTGTTCCCTTCACATT |
| orai1 | GATCGGCCAGAGTTACTCCG | TGGGTAGTCATGGTCTGTGTC |
| ltpr1 | GGGTCCTGCTCCACTTGAC | CCACATCTTGGCTAGTAACCAG |
| Atp2a2 | CTGTGGAGACCCTTGGTTGT | CAGAGCACAGATGGTGGCTA |
| Gck | ACCAAGCGGTATCAGCATGTG | TGGACTTCTCTGTGATTGGCA |
| Hk2 | ATGATCGCCTGCTTATTCACG | CGCCTAGAAATCTCCAGAAGGG |
| Hk3 | TGCTGCCCACATACGTGAG | GCCTGTCAGTGTTACCCACAA |
| Pkm | GGTGGCTCTGGATACAAAGGG | CACACTTCTCCATGTAAGCGT |
| Gpi | CTCAAGCTGCGCGAACTTTTT | GGTTCTTGGAGTAGTCCACCAG |
| Pgm1 | CAGAACCCTTTAACCTCTGAGTC | TCATTCATTCGAGAAATCCCTGC |
| Pgm2 | GCGGAATGGGATGAACAAGGA | GGTCATTGATGTAGCAAAACCCT |
| Pck1 | CTGCATAACGGTCTGGACTTC | GCCTTCCACGAACTTCCTCAC |
| G6pc | CTGAGCGCGGGCATCATAAT | GATTCTTAGGATCGCCCAGAAAG |
| Pgc1a | CCC TGC CAT TGT TAA GAC C | TGC TGC TGT TCC TGT TTT C |
| Ppara | TATTCGGCTGAAGCTGGTGTA | CTGGCATTTGTTCCGGTTCT |
| Cpt1 | GCTGGAGGTGGCTTTGGT | GCTTGGCGGATGTGGTTC |
| Slc25a20 | AGTCGGACCTTGACCGTGT | GACGAGCCGAAACCCATCAG |
| Scd1 | TTC TTG CGA TAC ACT CTG GTG C | CGG GAT GGA ATG TTC TTG TCG T |
| Fas | GGA GGT GGT GATA GCC GG TAT | TGG GTA ATC CATA GAG CCC AG |
| Serbpc1 | GGAGCCATGGATTGCACATT | GGCCCGGGAAGTCACTGT |
| 18S | AGTCCCTGCCCTTTGTACACA | CGATCCGAGGGCCTCACTA |

**Oligonucleotides for shRNA**

| stim1 | CCGGCCGAAACATCCATAAGCTGATCTCGA GATCAGCTTATGGATGTTTCGGTTTTTTG | TRCN0000193877 |
| --- | --- | --- |
| stim2 | CCGGGACGAAGTAGACCACAAGATTCTCG AGAATCTTGTGGTCTACTTCGTCTTTTTTG | TRCN0000187841 |
| ogt | CCGGCCCATTTCTTTCAGCAGAAATCTCGA GATTTCTGCTGAAAGAAATGGGTTTTTG | TRCN0000110395 |

DOI: https://doi.org/10.7554/eLife.29968.027

increased until ~7 mL/min afterwards. When the entire buffer I had been infused, it was substituted for 50 mL of buffer II. The buffers should be kept at ~37°C during the entire procedure. After perfusion, the primary hepatocytes were carefully released and sediment at 500 rpm for 2 min, washed two times and suspended with Williams E medium supplemented with 5% CCS and 1 mM glutamine (Invitrogen, CA). To separate live from dead cells, the solution of hepatocytes was layered on a 30% Percoll gradient and centrifuged ~1500 rpm for 10–15 min. The healthy cells were recovered at the bottom of the tube and plated for experimentation. Buffer I contained: 11 mM Glucose; 200 µM EGTA; 1.17 mM $MgSO_4$ heptahydrated; 1.19 mM $KH_2PO_4$; 118 mM NaCl; 4.7 mM KCl; 25 mM $NaHCO_3$, pH 7.32. Buffer II contained: 11 mM Glucose; 2.55 mM $CaCl_2$; 1.17 mM $MgSO_4$ heptahydrated; 1.19 mM $KH_2PO_4$; 118 mM NaCl; 4.7 mM KCl; 25 mM $NaHCO_3$; BSA (fatty acid free) 7.2 mg/mL; 0.18 mg/mL of Type IV Collagenase (Worthington Biochem Catalog: LS004188). BSA and collagenase were added immediately before use.

## Glucose production assay

Primary hepatocytes derived from stim1$^{fl/fl}$ and stim1$\Delta^{LIVER}$ animals kept on high fat diet were isolated and incubated in fresh William's E medium with 5% fetal bovine serum (FBS) for 4 hr and thereafter, incubated in 0.1% FBS overnight. Next day, cells were washed twice and incubated with DMEM without glucose (Invitrogen) supplemented with 10 mM HEPES in the presence of 2 mM pyruvate, 2 mM glutamate and 20 mM glycerol for 4 hr. The glucose concentrations in the media were determined by Amplex Red glucose/glucose oxidase assay system (Invitrogen).

## Total protein extraction, immunoprecipitation and western blotting

Liver tissues were homogenized in cold lysis buffer containing 50 mM Tris-HCl (pH 7.4), 2 mM EGTA, 5 mM EDTA, 30 mM NaF, 10 mM Na$_3$VO$_4$, 10 mM Na$_4$P$_2$O$_7$, 40 mM glycerophosphate, 1 % NP-40, and 1% protease inhibitor cocktail. After ~20 min incubation under mild agitation in the cold, the homogenates were centrifuged for 15 min at 9000 rpm to pellet cell debris, and protein concentrations were determined by BCA. Finally, the samples were diluted in 5x Laemmli buffer and boiled for 5 min. The protein lysates were subjected to SDS-polyacrylamide gel electrophoresis, as previously described (Arruda et al., 2014; Fu et al., 2012). The antibodies used in this study are listed in Table 2. All the immunoblots were incubated with primary antibody overnight at 4°C, followed by incubation with secondary antibody conjugated with horseradish peroxidase (Amersham Biosciences) for 1–3 hr at room temperature. Individual membranes were visualized using the enhanced chemiluminescence system (Roche Diagnostics). Immunoprecipitation (IP): For the immunoprecipitation studies, primary antibodies against STIM1 and O-GlcNAc were cross-linked on to protein-A magnetic beads with Dimethylpimelimidate (DMP). Cells lysates were prepared in a IP lysis buffer containing 50 mM Tris-HCl (pH 7.4), 1 mM EDTA, 150 mM NaCl, 10 mM Na$_3$VO$_4$, 1 % NP-40, and 1% protease inhibitor cocktail and 10 µM of PugNac. Protein content was adjusted to be the same in all samples and 300–400 µg of protein were incubated with the beads overnight. The next day, samples were washed 3x with IP buffer and eluted in Laemmli buffer and boiled for 5 min. Immunoblotting was performed as described above. For the IP of Flag tagged STIM1, the lysates were incubated with anti-flag M2 magnetic beads from Sigma and eluted by competition with 3x flag peptide.

## Gene expression analysis

Tissues in TRIzol (Invitrogen) were disrupted using TissueLyser (Qiagen). To obtain RNA, trizol homogenates were mixed with chloroform vortexed thoroughly and centrifuged 12000 g for 20 min at 4°C. The top layer was transferred to another tube and mixed with isopropanol and centrifuged again at 12000 g for 20 min at 4°C. The RNA found in the precipitate was washed twice with 70% Ethanol and diluted in RNAse free water. Complementary DNA was synthesized using iScript RT Supermix kit (Biorad). Quantitative real-time PCR reactions were performed in duplicates or triplicates on a ViiA7 system (Applied Biosystems) using SYBR green and custom primer sets or primer sets based on Harvard primer bank. Gene of interest cycle thresholds (Cts) were normalized to *18S ribosomal RNA (Rn18s)* house keeper levels by the ΔΔCt. The primers used for qPCR are listed in *Table 1*.

## Endogenous protein staining and confocal imaging

For thapsigargin (Tg) stimulation, primary hepatocytes or Hepa1-6 cells were seeded on 3.5 mm round glass dishes (1.5 mm) in Williams Medium in the presence of 5% CCS overnight at 37°C, 5% CO$_2$. The following morning, cells were washed, and in the afternoon they were incubated for a few minutes in Ca$^{2+}$-free medium (described in the Ca$^{2+}$ imaging section) followed by 1 µM Tg treatment (or vehicle, DMSO) for the duration indicated in the figure legends. Cells were fixed with 4% paraformaldehyde for 10 min at room temperature (RT) and washed 3x in PBS, before a 20 min permeabilization using 0.2% Triton-X100 in 2% BSA at RT. Primary antibodies were diluted 1:200 for STIM1 (Cell Signaling #4916), 1:100 for STIM2 (Cell Signaling #4917), for Na$^+$/K$^+$ ATPase $\alpha-1$ -Alexa Fluor 488 (Sigma #16–243) and for Anti-KDEL antibody, Alexa fluor 488 (Abcam, ab184819) in PBS and the cells were incubated in this solution over night at 4°C in the dark. The next day, cells were washed 3x, including one long wash for more than 10 min. Secondary antibody was diluted 1:2500 in PBS, and the cells were incubated with it at RT for 1 hr in the dark. The cells were washed 3x, including one long wash, and if needed, Hoechst was used as nuclear marker, diluted 1:1000 in PBS

**Table 2.** List of antibodies used in this study.

**Antibodies used in western blots**

| Antibody | Vendor | Catalog number |
|---|---|---|
| STIM1 | Cell signaling | #4916 |
| STIM2 | Cell signaling | #4917 |
| ORAI1 | Sigma | SAB3500126 |
| SERCA2b | Cell Signaling | #4388 |
| pJNK | Cell signaling | #81E11 |
| peIF2alpha | Cell signaling | #3597 |
| CHOP | Cell Signaling | mab# 2895 |
| pAKT | Santa Cruz | 7985R |
| AKT | Santa Cruz | 8312 |
| IR | Santa Cruz | 711 |
| pIR | Calbiochem | 407707 |
| OGT | Cell Signaling | #5368 |
| O-GlcNAc (CTD110.6) | Cell Signaling | #12938 |
| Flag M2 | Sigma | SLBD9930 |
| pCamKII | Cell Signaling | #12716 |
| β-tubulin | Abcam | ab 21058 |
| pSTIM1 | provided by Dr. Martin-Romero, University of Extremadura, Spain | |

DOI: https://doi.org/10.7554/eLife.29968.028

for 10 min at RT. For the experiments described in *Figure 1—figure supplement 1D*, *Figure 1—figure supplement 2A,B* cells were imaged at the Confocal and Light Microscopy Core Facility at the Dana Farber Cancer Institute with a Yokogawa CSU-X1 spinning disk confocal system (Andor Technology, South Windsor, CT) with a Nikon Ti-E inverted microscope (Nikon Instruments, Melville, NY), using a 60x or 100x Plan Apo objective lens with a Hamamatsu OrcaER camera. Andor iQ software (Andor Technology, South Windsor, CT) was used for acquisition parameters, shutters, filter positions and focus control. For *Figures 1C, D*, *3G* and *5D*, cells were imaged at the Harvard Medical School Imaging Core with a Yokogawa CSU-X1 spinning disk confocal system (Andor Technology, South Windsor, CT) with an iXon EMCCD camera and Metamorph was the software used for acquisition parameters, shutters, filter positions and focus control. For *Figure 2H* cells were image with a Yokogawa CSU-X1 spinning disk confocal system (Andor Technology, South Windsor, CT) with a Nikon Ti-E inverted microscope (Nikon Instruments, Melville, NY), using a 60x or 100x Plan Apo objective lens with Zyla cMOS camera and NIS elements software was used for acquisition parameters, shutters, filter positions and focus control.

## Confocal image analysis

Image analysis was performed using Fiji software. All the images in the same experiment were analyzed using the same parameters. For *Figure 1E*, puncta number was counted using a custom made macro. After background subtraction of 15, images were subject to 'Filter Gaussian blur' of 1.00 and a threshold was set to define puncta from background signal. In order to localize the individual puncta 'Find Maxima' tool was used and puncta were counted using 'Analyze Particles' tool. For *Figure 2J*, images were background subtracted and single cells or groups of cells either expressing OGT or not (identified by RFP expression), were cropped from the original image. Puncta above a threshold of 50 were counted using 'Analyze Particles', and normalized to number of cells in the field. Puncta pixel intensities: For *Figure 1F* and *Figure 1—figure supplement 2B*, a 125 × 125 pixels box was placed in a representative region of the cell and pixel intensities were recorded in the entire box using a custom made macro. Intensities were distributed into a histogram with R, and plotted in Prism. No further normalization was performed, as the same number of pixels was evaluated in all the images.

Plot profile in *Figures 1G*, *2L* and *5E*: A thin box of 40 × 250 pixels was placed across the cell and the average intensity contained in the box was plotted as a profile with Fiji. Edge/cyto measurements: for *Figures 1H*, *2N,5D*, *Figure 1—figure supplement 2B*: A small box was hand-drawn at a region of the edge of the cell (at the PM), and the same size box was placed right next to the edge, in the cytoplasmic region. The mean STIM signal across each box was recorded, and the ratio between the reading at the edge of the cell and in the cytoplasm was derived after background signal subtraction. Such pairs of boxes were drawn in four representative places in each cell, from 2 to 5 cells per field. Areas were chosen only on cell membranes containing neighbor cells.

## Total internal reflection fluorescence (TIRF) microscopy

For the TIRF images, cells were stained as described above using anti-STIM1 antibody and anti-$Na^+/K^+$ ATPase $\alpha-1$, Alexa Fluor 488 antibody to stain the PM. In each experiment epifluorescence images were acquired and the PM marker was used to define the PM area for the TIRF imaging. The images were acquired a Zeiss Elyra PS.1 microscope with a 60 or 100 × 1.46 N.A. oil immersion TIRF objective (Carl Zeiss GmbH, Germany) in TIRF mode with an Andor EM-CCD camera (512 × 512 pixels) at the Harvard center for Biological Imaging, Cambridge, MA.

## Cytosolic $Ca^{2+}$ imaging using Fura-2 AM and Fluo-4

Cells were loaded with 4 μM Fura-2AM and 1 μM Pluronic F-127 in HBSS for ~60 min at room temperature. Before imaging, the cells were washed and kept in a medium containing 10 mM Hepes, 150 mM NaCl, 4 mM KCl, 2 mM $CaCl_2$, 1 mM $MgCl_2$, 10 mM D-glucose, pH 7.4. $Ca^{2+}$-free medium was prepared similarly to the buffer described above, in the absence of $CaCl_2$ and in the presence of 2 mM EGTA and 3 mM $MgCl_2$. Cells were stimulated with 1 μM Tg for indirect ER $Ca^{2+}$ measurements, and in some cases supplemented with 100 μM 2-APB for SOCE inhibition. Ratiometric Fura-2AM imaging was performed by alternatively illuminated with 340 and 380 nm light for 250 ms (Lambda DG-4; Sutter Instrument Co.), using an Olympus IX70 with 40x objective. Emission light >510 nm was captured using a CCD camera (Orca-ER; Hamamatsu Photonics). Both channels were collected every 5 s, background corrected and analyzed with Slidebook and custom-made R-scripts. For the Fluo-4 experiments described in *Figure 2J*, the same procedure described above as used except that the cells were incubate with Fluo-4 for 30 min to 1 hr. The images were acquired on a Leica SP8X and LAS X software was used for acquisition parameters, shutters, filter positions and focus control.

## Cell culture

Loss of function experiments were performed in Hepa1-6 cells (a mouse hepatocyte cell line from ATCC, derivative of the BW7756 mouse hepatoma that arose in C57/L mouse) for STIM1, STIM2 and ORAI1. This cell line was negative for mycoplasma testing. Five different shRNA sequences targeting different regions of each gene were obtained from *MISSION* shRNA Library (Sigma) along with the Scramble shRNA control. Stable cell lines were generated by lentiviral infection of Hepa1-6 cells overnight in media (DMEM and 10% CCS and polybrene). After infection cells were incubated in growth media (DMEM and 10% CCS) and selected with 3 μg/μl puromycine. The knockdown efficiency of each shRNA was evaluated by western blot analysis and the experiments were done with the sequence that generated the highest efficiency. Cells were frozen at passage two after selection and fresh cells were thawed for each experiment. The sequences of the shRNA used in the study are listed in *Table 1*. For the overexpression of OGT, a hOGT/RFP co-expression plasmid was generated using a self-cleaving P2A peptide linker. RFP was first cloned into pcDNA6 Myc/His B (Life Technologies V22120) at BamHI/XhoI sites. OGT-P2A was generated by PCR (F: 5'-CCC GGT ACC GCC ACC ATG GCGTCTTCCGTGGGCAA; R: 5'- CCC GGA TCC AGG TCC AGG GTT CTC CTC CAC GTC TCC AGC CTG CTT CAG CAG GCT GAA GTT AGT AGC TCC GCT TCC TGCTGACTCAGTGACTTC) and cloned into the above vector at KpnI/BamHI sites.The OGT-RFP plasmid was transfected into, cells using lipofectamine 2000 overnight in OptiMEM media and the experiment done after 36–48 hr after transfection. For the STIM1-flag pull downs, 3xFLAG-mSTIM1 was generated by PCR (F: 5'-CCC GGATCC GCC ACC **ATG** *GAC TAC AAA GAC CAT GAC GGT GAT TAT AAA GAT CAT GAC ATC GAT TAC AAG GAT GAC GAT GAC AAG* GATGTGTGCGCCCGTCTTGCCCTGT; R: 5'- CCC GCGGCCGC TTA CTA CTTCTTAAGAGGCTTCTTAA) and cloned into pENTR1A no CCDB vector

(Addgene #17398) at BamHI/NotI sites and then transferred to pLenti CMV Puro DEST (w118-1) (Addgene # 17452) using Gateway LR Clonase II Enzyme Mix (Life Technologies 11791). STIM1-flag was transfected in HEK293T cells and after 24–36 hours cells were trated with 3 ug/mL of puromycin for selection and a stable cell line was generated.

## Insulin treatment in vivo and in vitro and lipid loading

For insulin signaling studies, cells were serum deprived for 4 hr before stimulus with 3 nM of insulin for 3 and 6 min. Cells were washed in ice-cold PBS and snap-frozen in liquid nitrogen. For the insulin infusions in vivo, following 6 hr of food withdrawal, mice were anaesthetized with an intraperitoneal injection of 2 mg/ml xylazine and 2 mg/ml ketamine, and insulin (0.75 IU kg$^{-1}$) or phosphate buffered saline (PBS) in 200 μl volume was infused into the portal vein. Three minutes after infusion, tissues were removed and frozen in liquid nitrogen and kept at –80°C until processing. For lipid loading experiments primary hepatocytes were isolated in the morning and kept in Williams E medium. After the cells were settled, Williams medium containing 1 mM Oleic acid and 40 μM palmitic acid was added to the culture. Cells before and after 16 hr of lipid loading and stained with BODIPY and Hoechst to visualize lipid droplets and nucleus, respectively.

## Transmission electron microscopy (TEM)

For TEM, primary hepatocytes were fixed in a 1:1 ratio in a fixative buffer and Williams Medium. The fixative buffer contained: 4 parts of FP stock (2.5% PFA, 0.06% picric acid in 0.2M Sodium Cacodylate buffer pH 7.4) and 1 part of 25% glutaraldehyde for at least 2 hr. The samples were then placed in propyleneoxide for 1 hr and infiltrated in a 1:1 mixture of propyleneoxide and TAAB 812 Resin mixture (Marivac Canada Inc. St. Laurent, Canada). Sectioning and imaging: ultrathin sections (about 90 nm) were generated using a Reichert Ultracut-S microtome, JEOL microscope. Images were recorded with an AMT 2 k CCD camera. For the STIM-HRP labeled cells, primary hepatocytes were infected with adenovirus expressing STIM1-HRP. The construct was kindly provided by Dr. Lewis from Stanford University and cloned in an adenovirus (serotype 5, Ad5). Primary hepatocytes were fixed in a buffer containing 2% glutaraldehyde in 0.1 M sodium cacodylate at room temperature for 1 hr, then quickly moved to ice and washed with 0.1 M sodium cacodylate buffer for 10 min. HRP was visualized with 0.5 mg ml diaminobenzidine and 0.03% hydrogen peroxide in 0.1 M sodium cacodylate buffer for 5 min-20min. The reaction was stopped by five washes with cold 0.1 M sodium cacodylate buffer.

## Acknowledgements

We are grateful to D Clapham and N Blair for their help in the Ca2 +imaging experiments and generously allowing the use of their laboratory facilities. We thank E. Benecchi for their assistance in electron microscopy. We acknowledge an NIH SIG award (1S10RR029237-01, to Samuel Kunes), which was used to acquire the ELYRA microscope used for the TIRF experiments. Antibodies to detect pSTIM1 were kindly provided by Dr. Martin-Romero from University of Extremadura, Spain. pCMV-hOGT and adenovirus expressing shRNA against OGT were a gift from Dr. Yang from Yale University, USA. We would like to thank Dr. Rao from La Jolla Institute for Allergy and Immunology for kindly providing us the stim1$^{fl/fl}$ mice. STIM1-HRP was a gift from Dr. Lewis from Stanford University. We would like to give a special thanks to Dr. Claiborn and Dr. Bartelt for critical reading and editing of the manuscript. This work was supported by the US National Institutes of Health (DK52539 to GSH) BMP was supported by Alfred Benzon Foundation and the Lundbeck Foundation (Denmark). APA was supported by PEW Charitable Trusts.

## Additional information

### Funding

| Funder | Grant reference number | Author |
|---|---|---|
| National Institutes of Health | DK52539 | Gökhan S Hotamisligil |
| Pew Charitable Trusts | | Ana Paula Arruda |

| Alfred Benzon Foundation | Benedicte Mengel Pers |
| --- | --- |
| Lundbeckfonden | Benedicte Mengel Pers |

The funders had no role in study design, data collection and interpretation, or the decision to submit the work for publication.

## Author contributions
Ana Paula Arruda, Conceptualization, Formal analysis, Validation, Investigation, Visualization, Methodology, Writing—review and editing; Benedicte Mengel Pers, Conceptualization, Formal analysis, Validation, Investigation, Visualization, Methodology, Writing—original draft, Writing—review and editing; Günes Parlakgul, Ekin Güney, Formal analysis, Investigation, Visualization, Methodology, Writing—review and editing; Ted Goh, Formal analysis, Investigation; Erika Cagampan, Grace Yankun Lee, Renata L Goncalves, Formal analysis, Investigation, Writing—review and editing; Gökhan S Hotamisligil, Conceptualization, Resources, Supervision, Funding acquisition, Writing—original draft, Project administration, Writing—review and editing

## Author ORCIDs
Ana Paula Arruda (iD) http://orcid.org/0000-0001-6179-2687
Gökhan S Hotamisligil (iD) http://orcid.org/0000-0003-2906-1897

## Ethics
Animal experimentation: All in vivo studies are approved by the Harvard Medical Area Standing Committee on Animals under the protocols #02396 and #04779.

## Decision letter and Author response
Decision letter https://doi.org/10.7554/eLife.29968.032
Author response https://doi.org/10.7554/eLife.29968.033

## Additional files
### Supplementary files
• Supplementary file 1. Source uncropped western blots: this file contains all the un-cropped western blot images presented in this manuscript.
DOI: https://doi.org/10.7554/eLife.29968.029

• Transparent reporting form
DOI: https://doi.org/10.7554/eLife.29968.030

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
