## [Decision Letter]

[Editors’ note: this article was originally rejected after discussions between the reviewers, but the authors were invited to resubmit after an appeal against the decision.]

Thank you for submitting your work entitled "Defective STIM-mediated Store Operated Ca^2+^ Entry in Hepatocytes Leads to Metabolic Dysfunction in Obesity" for consideration by *eLife*. Your article has been reviewed by three peer reviewers, one of whom is a member of our Board of Reviewing Editors, and the evaluation has been overseen by a Senior Editor. The reviewers have opted to remain anonymous.

Our decision has been reached after consultation between the reviewers. Based on these discussions and the individual reviews below, we regret to inform you that your work will not be considered further for publication in *eLife*.

It was agreed that the manuscript was very well crafted and written and the fundamental observation that SOCE is important for hepatic calcium homeostasis and glucose tolerance is solid. However, there are many basic questions remaining and the underlying mechanisms have not yet been clearly addressed. The reviewers raised many basic issues and on balance many of these would have to be clarified to raise the level of scientific impact to the point where the paper could be considered for *eLife*.

*Reviewer #1:*

This manuscript describes a novel connection between obesity associated glucose intolerance and hepatic calcium transport into the ER mediated by the SOCE system. The concept that glucose intolerance in obesity is promoted by dysfunction of STIM mediated by O-glycNAcylation is of significant interest. However, there are several key points that the authors should consider in order to more completely document and strengthen this hypothesis:

1) It is not clear that the glucose intolerance (abnormal GTT) induced by the manipulations (STIM KO and over expression in liver) relates to systemic insulin resistance versus alterations in insulin, glucagon or other hormones during the GTT. Serum hormones and hyperinsulinemic clamp studies must be carried out to show that STIM KO suppresses insulin mediated inhibition of hepatic glucose output in vivo. The limited data indicates some reduction in insulin stimulated Akt, but it is not clear whether this translates to actual glucose production.

2) The authors suggest that overnutrition is associated with hyper-glyNAcylated proteins as is hyperglycemia. If the hyper-glyNAcylated STIM1 is promoted by hyperglycemia, then this is a secondary effect of the obesity and insulin resistance and not the cause it would seem. How early in the hyperglycemic stage of the ob/ob mouse is STIM1 translocation observed to be defective? Does it only occur well after the extreme hyperinsulinemia and hyperglycemia is established in these ob/ob animals? Does it occur early or late in HFD mice?

3) The KO and overexpression of STIM1 in liver don't mimic alterations in the obese state, since STIM1 expression is not altered in the ob/ob mouse, but rather its translocation. Under the overexpression conditions, one might predict that STIM1 is also hyper-glyNAcylated and therefore also defective. Is that the case, or does it escape this modification when highly expressed? In order to document that hyper-glyNAcylated STIM1 is the cause of altered calcium dynamics in the obese state, one would have to specifically block this modification only in STIM1, a difficult experimental feat. Thus, the hypothesis at this point is rather weakly supported as it is "guilt by association" rather than demonstrated causation.

Reviewer #2:

The authors present an interesting line of evidence that describes Ca^2+^ handling as a key mechanism underlying hepatic endoplasmic reticulum (ER) stress in obesity. They describe that hepatocytes from obese mice displayed significantly diminished store operated Ca^2+^ entry (SOCE) as a result of impaired STIM1 translocation, which was associated with aberrant STIM1 O-GlycNAcylation.

Primary hepatocytes deficient in STIM1 exhibited elevated cellular stress as well as impaired insulin action and increased lipid droplet accumulation and mice with acute liver deletion of STIM1 displayed systemic glucose intolerance. Conversely, over-expression of STIM1 in obese mice led to increased SOCE, which was sufficient to improve systemic glucose tolerance.

Reviewer Synopsis:

Overall the study is of potential significant interest to the field and the experiments described are well-performed. The major conceptual advance is that Ca^2+^ balance regulation by SOCE in hepatocytes contributes to metabolic homeostasis. Major comments are listed below:

1) How does Ca^2+^ balance regulates lipid accumulation in hepatocytes?

In Figure 3, the authors show that STIM1 deficiency leads to accumulation of lipids following treatment with palmitic and oleic acid in vitro. Is the increased accumulation of lipids in hepatocytes due to the increase entry of fatty acids or a decreased fatty acid oxidation? in vivo, there was no increase in lipid accumulation in liver of mice with hepatocyte-specific deletion of STIM1. This could be due to a compensatory increase in STIM2. Therefore, silencing both STIM1 and 2 could reproduce in vivo the increased lipid accumulation in liver. The authors could use shRNA adenovirus targeting STIM1/2 and measure lipid accumulation. On the other hand, it could be due to the fact that the mice used in the experiment were fed a high fat diet and already display hepatic steatosis, which would mask the effect of STIM1 deficiency. Using mice fed a normal diet with deletion of STIM1 would be a better model to measure an increased lipid accumulation and glucose intolerance exacerbation.

2) In Figure 4, STIM1 overexpression in ob/ob mice improves glucose tolerance and insulin response, and decreases expression of gluconeogenic genes while increasing the expression of genes involved in glycolysis. What is the effect of STIM1 overexpression on hepatic steatosis: is there more TG accumulation in liver? Is the expression of genes involved in de novo lipogenesis, lipid uptake, FA oxidation affected by STIM1 overexpression?

Reviewer #3:

Pers and colleagues present data linking obesity with defective calcium influx and endoplasmic reticulum dysfunction in mouse hepatocytes. In general, the manuscript is well written and the experimental logic is sound. The authors conclude that STIM1 trafficking from the ER to plasma membrane is perturbed in hepatocytes from obese mice due to aberrant/increased O-GycNAcylation. This coincides with increased ER stress and a variety of metabolic defects that are interesting but not at all surprising considering the very well-documented connection between ER calcium homeostasis and ER function. The most exciting portion of the manuscript is the demonstration that over-expression of STIM1-YFP rescues several of the metabolic defects observed in obese mice.

Unfortunately, it is not at all clear why this is the case or if it even involves increased STIM1 activity/SOCE in the obese hepatocytes. The data indicate that STIM1-YFP overexpression severely perturbs or alters ER morphology potentially allowing STIM1 to interact with Orai1 at the plasma membrane, essentially by chance. However, STIM1-YFP appears to be quite different than endogenous STIM1 as it is not affected by thapsigargin. Many questions remain: Is overexpressed STIM1 also O-GlcNAc-ylated? Is ER calcium homeostasis improved? Does over-expression of any ER-localized transmembrane protein provide relief in obese mouse hepatocytes? Why does overexpression of STIM1 result in increased transcription of Orai1 mRNA, SERCA mRNA? STIM1 overexpression may simply activate a compensatory response to ER dysfunction such as the UPR which improves the documented metabolic defects. Also, why/how does STIM1 overexpression increase glycolysis gene expression?

While the data is beautifully presented, the impact of the paper's findings are modest at best. The conclusion that SOCE is an important mechanism for healthy hepatocyte Ca^2+^ homeostasis and systemic metabolic control is of moderate interest.

---

## [Author Response]

[Editors’ note: the author responses to the first round of peer review follow.]

Reviewer #1:This manuscript describes a novel connection between obesity associated glucose intolerance and hepatic calcium transport into the ER mediated by the SOCE system. The concept that glucose intolerance in obesity is promoted by dysfunction of STIM mediated by O-glycNAcylation is of significant interest. However, there are several key points that the authors should consider in order to more completely document and strengthen this hypothesis:1) It is not clear that the glucose intolerance (abnormal GTT) induced by the manipulations (STIM KO and over expression in liver) relates to systemic insulin resistance versus alterations in insulin, glucagon or other hormones during the GTT. Serum hormones and hyperinsulinemic clamp studies must be carried out to show that STIM KO suppresses insulin mediated inhibition of hepatic glucose output in vivo. The limited data indicates some reduction in insulin stimulated Akt, but it is not clear whether this translates to actual glucose production.

Thank you for this thoughtful question. To address this issue, we measured fasting insulin levels in mice in which STIM1 was down regulated (STIM1KD), and found no differences compared with the WT controls (new Figure 3). We had very limited serum samples from these animals and unfortunately were not able to determine other hormones during GTT. However, we decided to directly determine whether the absence of STIM1 influences hepatocyte glucose production, by measuring glucose production in primary hepatocytes isolated from WT and STIM1 deficient animals. As shown in the new Figure 3, glucose production stimulated by gluconeogenic substrates (pyruvate, glutamine and glycerol) was significantly higher in STIM1 deficient animals compared with WT. These data indicate that the effect of STIM1 deficiency in hepatocytes does in deed translate to higher glucose production.

While we agree that hyperinsulinemic clamps are the gold standard method for determination of glucose fluxes, it is not possible to perform these experiments in a timely manner considering the complexity and time demands. However, our new results showing significantly increased glucose production from primary hepatocytes with STIM1-deficiency add an additional and direct measure from hepatocytes to support our conclusions.

2) The authors suggest that overnutrition is associated with hyper-glyNAcylated proteins as is hyperglycemia. If the hyper-glyNAcylated STIM1 is promoted by hyperglycemia, then this is a secondary effect of the obesity and insulin resistance and not the cause it would seem. How early in the hyperglycemic stage of the ob/ob mouse is STIM1 translocation observed to be defective? Does it only occur well after the extreme hyperinsulinemia and hyperglycemia is established in these ob/ob animals? Does it occur early or late in HFD mice?

We thank the reviewer for these comments. To answer this question, we followed a cohort of animals fed chow or HFD for 3, 5, 7 and 11 weeks. We choose to perform the experiment using HFD obesity model instead of the ob/ob because in leptin deficient animals, the hyperglycemia and hyperinsulinemia appears very early on (as early as 4 weeks) which would make it difficult to uncouple the STIM defects from the acute metabolic alterations found in this mouse model.

As can be seen in Figure 1—figure supplement 2, in HFD fed mice, body weight and insulin levels started to rise after 5 weeks of high fat feeding, reaching maximal levels after 7 weeks of HFD. However, the glucose levels were not significantly changed during this time frame.

We then analyzed STIM1 translocation in hepatocytes isolated from these mice. As shown in Figure 1—figure supplement 2, at 3 weeks time point on HFD we detected some degree of translocation at the baseline (non-treated condition) indicating that short term HFD is sufficient to induce decreased levels of calcium in the ER, which is the driving force for STIM translocation. However, at this time point, STIM1 translocation upon TG stimulation was not significantly impaired compared with cells derived from chow fed animals. The defect in TG ‐induced STIM1 translocation was first observed in hepatocytes isolated from mice after 5 weeks of HFD, and was more pronounced at the 7 and 11-week timepoints (Figure 1—figure supplement 2). Based on these findings, we conclude that the defect in STIM1 translocation is not related to hyperglycemia and starts before the frank hyper-insulinemia found in obesity at least in the HFD model Figure 1—figure supplement 2. We additionally observed that the O-GlcNAcylation levels of STIM1 are progressively increased in hepatocytes from HFD fed animals starting from 3 week on HFD (Figure 2—figure supplement 1).

3) The KO and overexpression of STIM1 in liver don't mimic alterations in the obese state, since STIM1 expression is not altered in the ob/ob mouse, but rather its translocation. Under the overexpression conditions, one might predict that STIM1 is also hyper-glyNAcylated and therefore also defective. Is that the case, or does it escape this modification when highly expressed? In order to document that hyper-glyNAcylated STIM1 is the cause of altered calcium dynamics in the obese state, one would have to specifically block this modification only in STIM1, a difficult experimental feat. Thus, the hypothesis at this point is rather weakly supported as it is "guilt by association" rather than demonstrated causation.

To answer this reviewer’s question we used adenovirus-mediated gene delivery to overexpress STIM1-YFP in primary hepatocytes derived from obese (ob/ob) animals. We immunoprecipitated O-GlcNac modified proteins in these cells using an OglcNac specific antibody and performed immunoblot to detect STIM1 among these proteins. As it can be seen in Figure 4, STIM1-‐YFP overexpression resulted in an approximately 20-fold increase in the expression of STIM1 protein. However, the amount of STIM1 observed in the O-‐GlcNac pull-‐down was not proportionally increased, indicating that a majority of the overexpressed STIM1 is able to “escape” this post translation modification.

So far, we have not been able to make satisfactory progress in determining the exact sites of STIM1 O-GlcNacylation in order to perform site direct mutagenesis. However, to determine if blocking this modification could rescue the defect in STIM1 translocation, we used an adenovirus expressing an shRNA against OGT to down-regulate OGT and therefore suppress O-GlcNacylation in primary hepatocytes from ob/ob animals. As shown in Figure 2, adenovirus mediated expression of shRNA against OGT lead to 60% down-regulation in OGT and overall OglcNacylation. Importantly, down-‐regulation of OGT lead to increased levels of STIM1 translocation in response Tg (Figure 2). These results strengthen our conclusion that O-GlcNacylation is an important factor driving the defects in STIM translocation in obese cells.

Reviewer #2:The authors present an interesting line of evidence that describes Ca^2+^ handling as a key mechanism underlying hepatic endoplasmic reticulum (ER) stress in obesity. They describe that hepatocytes from obese mice displayed significantly diminished store operated Ca^2+^ entry (SOCE) as a result of impaired STIM1 translocation, which was associated with aberrant STIM1 O-GlycNAcylation.Primary hepatocytes deficient in STIM1 exhibited elevated cellular stress as well as impaired insulin action and increased lipid droplet accumulation and mice with acute liver deletion of STIM1 displayed systemic glucose intolerance. Conversely, over-expression of STIM1 in obese mice led to increased SOCE, which was sufficient to improve systemic glucose tolerance.Reviewer Synopsis:Overall the study is of potential significant interest to the field and the experiments described are well-performed. The major conceptual advance is that Ca^2+^ balance regulation by SOCE in hepatocytes contributes to metabolic homeostasis. Major comments are listed below:1) How does Ca^2+^ balance regulates lipid accumulation in hepatocytes?In Figure 3, the authors show that STIM1 deficiency leads to accumulation of lipids following treatment with palmitic and oleic acid in vitro. Is the increased accumulation of lipids in hepatocytes due to the increase entry of fatty acids or a decreased fatty acid oxidation?

Thank you for this interesting question. The effects of STIM1 deficiency on lipid metabolism have been addressed recently by Maus et al., 2016. In this paper, the authors show that SOCE regulates lipolysis through the production of cyclic AMP and the expression lipases ATGL and HSL as well as the transcriptional regulators of fatty acid oxidation such as PGC-1a and PPARa. They also show, similar to us, that cells deficient in STIM1 accumulate lipid droplets. We have incorporated this reference into our discussion.

In vivo, there was no increase in lipid accumulation in liver of mice with hepatocyte-specific deletion of STIM1. This could be due to a compensatory increase in STIM2. Therefore, silencing both STIM1 and 2 could reproduce in vivo the increased lipid accumulation in liver. The authors could use shRNA adenovirus targeting STIM1/2 and measure lipid accumulation. On the other hand, it could be due to the fact that the mice used in the experiment were fed a high fat diet and already display hepatic steatosis, which would mask the effect of STIM1 deficiency. Using mice fed a normal diet with deletion of STIM1 would be a better model to measure an increased lipid accumulation and glucose intolerance exacerbation.

Thank you for this suggestion. As suggested, we have measured glucose tolerance and TG content in the livers of STIM1 deficient animals under chow diet and we were not able to detect significant differences between the genotypes on the overall TG content and glucose tolerance. We added a line referring to this data in the text. We also performed a short term HFD study (6 weeks on HFD) and observed that in this case STIM1 deficient animals exhibited a modest but significant, impairment in a glucose tolerance test compared with the WT controls. Additionally, these animals showed decreased insulin signaling as measured by pIR and pAKT. In this context there were no significant differences in Tg content, although a tendency to be higher was detected (Figure 3—figure supplement 2).

Interestingly, although we haven’t observed overall changes in lipid accumulation in the liver specific STIM1 deficient animals, we have noticed that these animals show higher content of microvesicular steatosis compared with controls where the majority of the lipid droplets are presented as large droplets (Figure 3—figure supplement 2). Although not addressed in this paper, previous reports have shown that microsteatosis are associated with mitochondrial dysfunction (Tandra et al., 2011), which would fit with the findings from Maus et al., 2016, showing that STIM deletion affects mitochondrial fatty acid oxidation.

As the reviewer’s pointed out, the interpretation of the phenotype of liver specific STIM1 deficient animals is complex. Under chow diet, the absence of a phenotype in STIM1 deficient animals under chow diet could possibly be due to a compensation of STIM2. As the animals progress on HFD, as shown in our time course analysis (Figure 1—figure supplement 2), STIM becomes deficient, so the comparison of the WT on HFD (STIM deficiency) with a KO on a HFD (STIM absence) is hard to interpret. We agree that possibly a knockdown of both STIM1 and STIM2 in the liver could provide answers for this matter, but given the time frame of this experiments, we opted to focus only on STIM1.

2) In Figure 4, STIM1 overexpression in ob/ob mice improves glucose tolerance and insulin response, and decreases expression of gluconeogenic genes while increasing the expression of genes involved in glycolysis. What is the effect of STIM1 overexpression on hepatic steatosis: is there more TG accumulation in liver? Is the expression of genes involved in de novo lipogenesis, lipid uptake, FA oxidation affected by STIM1 overexpression?

We have now measured TG content in liver from 3 different cohorts of ob/ob mice overexpressing STIM1-‐YFP and GFP control. As shown in Figure 5, TG content was significantly decreased in livers overexpressing STIM1 compared with controls. Accordingly, mRNA expression of CPT1 and acadvl1, genes involved fatty acid oxidation, are increased in STIM1-‐YFP overexpression livers compared with controls, while no changes were observed in genes involved in lipogenesis (Figure 5—figure supplement 1).

Reviewer #3:[…] The data indicate that STIM1-‐YFP overexpression severely perturbs or alters ER morphology potentially allowing STIM1 to interact with Orai1 at the plasma membrane, essentially by chance. However, STIM1-‐YFP appears to be quite different than endogenous STIM1 as it is not affected by thapsigargin.

We thank the reviewer for these comments. We agree that determining the response of the STIM1-YFP is a critical control and we have performed this experiment and demonstrated its proper responsiveness in the original manuscript (Figure 4). We recognize that our description of the STIM1-‐YFP experiments and the effect on SOCE may not have been sufficiently emphasized in the first version of the manuscript. As shown in Figure 4, and quantified in Figure 4, STIM1-YFP does respond to thapsigargin similarly to endogenous STIM1. To clarify this point further, we are now adding a live movie showing this response (Movie1) in the revised manuscript. Additionally, the STIM1-‐ YFP construct has been used multiple times in the literature, and shown to recapitulate the action of the endogenous protein (Sauc et al., 2015; Sundivakkam et al., 2012; Darbellay et al., 2010; Liou et al., 2005). The alterations in the ER morphology promoted by overexpression of STIM1 is likely related to the role of STIM1 as an ER structural protein. In resting cells STIM1 binds the plus-‐end-‐tracking protein (+TIP) of microtubules, anchoring the ER to the microtubules. However, to mediate SOCE, STIM1-YFP disconnects from the microtubules and needs to traffic within the ER, and this movement is driven by ER calcium depletion, phosphorylation and interaction with other proteins. To better clarify this process, we re-organize these data and its description accordingly in the revised manuscript.

Is overexpressed STIM1 also O-‐GlcNAcylated?

We thank the reviewer for this question. This issue was similarly raised by Reviewer 1. To address this reviewer’s question we used adenovirus-‐mediated gene delivery to express STIM1-‐YFP in primary hepatocytes derived from obese (ob/ob) animals. We immunoprecipitated O-GlcNac modified proteins in these cells using an O-GlcNac specific antibody and performed immunoblot experiments to detect STIM1 among these proteins. As can be seen in Figure 4, STIM1-YFP overexpression resulted in an approximately 20-fold increase in the expression of STIM1 protein. However, the amount of STIM1 observed in the O-GlcNac pull-down was not proportionally increased, indicating that a majority of the overexpressed STIM1 is able to “escape” this post translational modification.

Is ER calcium homeostasis improved?

As shown in Figure 4in the revised manuscript, overexpression of STIM1 leads to increased ER Ca^2+^content and SOCE. Does over-‐expression of any ER-localized transmembrane protein provides relief in obese mouse hepatocytes?

We would not anticipate that this would be the case. For example, in our earlier work (Yang et al., 2015) overexpression of IRE1, an ER trans-membrane protein, does not change the metabolic status of ob/ob animals while the overexpression of a mutant IRE1 resistant to nitrosylation does. Another example was reported recently by our group where the overexpression of the ER bound transcription factor Nrf1/Nfe2L1 revert steatosis in a model of hyperlipidemia/hypercholesterolemia, but overexpression of ER-bound transcription-inactive mutant versions of this protein does not affect steatosis (Widenmaier et al., 2017).

These are two clear examples that the relief is related to the function of the protein rather than an unspecific effect. In the current study, we have demonstrated that exogenously expressed STIM1 is functional and responds to calcium fluxes in a similar way to the endogenous protein. Taken together, we do not think that any ER-localized transmembrane protein is capable of generating the relief.

Why does overexpression of STIM1 result in increased transcription of Orai1 mRNA, SERCA mRNA?

Thank you for this interesting question. Abell et al., 2011 have shown experimentally and computationally that there is a tight regulation and adaptation between the different components of cellular calcium homeostasis; Knockdown of one of the players (i.e. SERCA, STIM1, PMCA, IP3R) will result in a compensatory regulation of the other proteins to maintain calcium homeostasis. Although Abell et al. did not directly assess the effect of overexpression of these calcium-‐regulating proteins, based on their knockdown experiments they formulated the hypothesis that an adaptive feedback loop exists between the components of the cellular calcium homeostasis. Hence, the increased SERCA and Orai levels in STIM1-‐OE hepatocytes, could likely be a compensatory response to restore cytosolic calcium levels which otherwise will be detrimental for the cell. We have incorporated explanation of this point into the discussion.

Also, why/how does STIM1 overexpression increase glycolysis gene expression?

It has been shown very recently that in T cells, SOCE controls the expression of glycolytic genes specifically through the regulation of the transcriptional factor NFAT (Vaeth et al., 2017). Although not experimentally addressed in our paper, a similar mechanism could exist in hepatocytes. We added a line of discussion about these findings in our manuscript.

References:

Tandra S, Yeh MM, Brunt EM, Vuppalanchi R, Cummings OW, Ünalp-‐Arida A, Wilson LA, Chalasani N. Presence and significance of microvesicular steatosis in nonalcoholic fatty liver disease. J Hepatol. 2011 Sep;55(3):654-‐659.

Saüc S, Bulla M, Nunes P, Orci L, Marchetti A, Antigny F, Bernheim L, Cosson P, Frieden M, Demaurex N. STIM1L traps and gates Orai1 channels without remodeling the cortical ER. J Cell Sci. 2015 Apr 15;128(8):1568-‐79.

Sundivakkam PC, Freichel M, Singh V, Yuan JP, Vogel SM, Flockerzi V, Malik AB, Tiruppathi C. The Ca(2+) sensor stromal interaction molecule 1 (STIM1) is necessary and sufficient for the store-‐operated Ca(2+) entry function of transient receptor potential canonical (TRPC) 1 and 4 channels in endothelial cells. Mol Pharmacol. 2012 Apr;81(4):510-‐26.

Darbellay B, Arnaudeau S, Bader CR, Konig S, Bernheim L. STIM1L is a new actin-‐ binding splice variant involved in fast repetitive Ca^2+^ release. J Cell Biol. 2011 Jul 25;194(2):335-‐46.

Yang L, Calay ES, Fan J, Arduini A, Kunz RC, Gygi SP, Yalcin A, Fu S, Hotamisligil GS. METABOLISM. S-‐Nitrosylation links obesity-‐associated inflammation to endoplasmic reticulum dysfunction. Science. 2015 Jul 31;349(6247):500-‐6.

Widenmaier SB, Snyder NA, Nguyen TB, Arduini A, Lee GY, Arruda AP, Saksi J, Bartelt A, Hotamisligil GS. NRF1 Is an ER Membrane Sensor that Is Central to Cholesterol Homeostasis. Cell. 2017 Nov 16;171(5):1094-‐1109.